# d2: Improved Techniques for Training Reasoning Diffusion Language Models

## Abstract

While diffusion language models (DLMs) have achieved competitive performance in text generation, improving their reasoning ability with reinforcement learning remains an active research area. Here, we introduce d2, a reasoning framework tailored for masked DLMs. Central to our framework is a new policy gradient algorithm that relies on properties of masking to accurately estimate the likelihoods of sampling trajectories. Our estimators trade off computation for approximation accuracy in an analytically tractable manner, and are particularly effective for DLMs that support any-order likelihood estimation. We characterize and study this property in popular DLMs and show that it is key for efficient diffusion-based reasoning. Empirically, d2 significantly improves over previous diffusion reasoning frameworks using only RL (without relying on supervised fine-tuning), and sets a new state-of-the-art performance for DLMs on logical reasoning tasks (Countdown and Sudoku) and math reasoning benchmarks (GSM8K and MATH500).

## 1 Introduction

Diffusion language models (DLMs) (Nie et al., 2025; Ye et al., 2025; Gong et al., 2025; Song et al., 2025) have recently emerged as a competitive alternative to autoregressive (AR) models for text generation, featuring attractive properties such as controllability (Nisonoff et al., 2024; Schiff et al., 2025) and fast parallel generation (Wang et al., 2025a; Khanna et al., 2025). Yet, while reinforcement learning (RL) has become the de-facto approach for inducing reasoning in AR LLMs, post-training of DLMs using RL remains an active research area.

Here, we introduce d2, a reasoning framework that improves over previous approaches in the setting of masked diffusion, a popular type of discrete diffusion. Central to our framework is a new policy gradient algorithm that relies on properties of masking for the efficient estimation of the likelihood of sampling trajectories. We complement this algorithm with practical recipes for post-training DLMs using RL that achieve high performance without relying on supervised fine-tuning, as well as a theoretical analysis.

Our technical approach starts from a policy gradient formulation in which likelihoods of trajectories serve as importance weights. We propose two estimators of these likelihoods. The first, d2-StepMerge, evaluates trajectories only at specific steps, reducing forward passes while controlling the approximation error; we theoretically study this error via a bound that decreases with the number of steps,

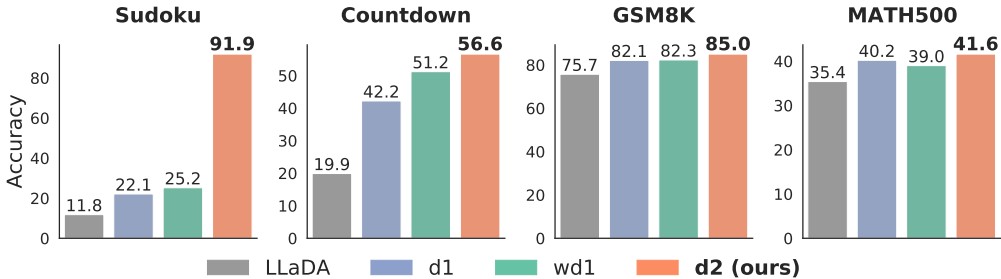

Figure 1: Benchmark performance of different RL post-training algorithms applied to LLaDA-8B-Instruct (Nie et al., 2025). Even without supervised finetuning (SFT), d2 outperforms d1 (Zhao et al., 2025) with SFT and wd1 (Tang et al., 2025) on all four reasoning benchmarks.

quantifying a compute-bias tradeoff. The second estimator, d2-AnyOrder, enables unbiased one-shot trajectory likelihood estimation under an any-order causal decoding property (Hoogeboom et al., 2021); we characterize this property and show that popular masked diffusion models such as LLaDA (Nie et al., 2025) do not satisfy it by default, motivating architectures that do. Our analysis provides evidence that enforcing any-order causality is key for efficient diffusion-based reasoning.

Empirically, d2 improves reasoning without supervised chain-of-thought (Wei et al., 2022) fine-tuning: applied to LLaDA-8B-Instruct, it surpasses prior diffusion-based RL methods on logical (Countdown, Sudoku) and mathematical (GSM8K, MATH500) benchmarks, establishing a new state of the art for DLMs under matched FLOPs. We further demonstrate the one-shot estimator in a red-teaming setup, where d2-AnyOrder efficiently steers an any-order causal DLM compared to DDPO (Black et al., 2023).

In summary, our work makes the following contributions: (1) We derive a GRPO-style RL policy gradient algorithm for DLMs that relies on trajectory likelihood estimates as importance weights. (2) We introduce and study two estimators computing these likelihoods: d2-StepMerge, which comes in analytical bounds on its approximation error, and d2-AnyOrder, which enables unbiased one-pass likelihood estimates under an any-order causal property that we characterize and study. (3) We complement our algorithm with a practical post-training recipe that achieves state-of-the-art reasoning results among DLMs on Sudoku, Countdown, GSM8K, and MATH500, without relying on supervised fine-tuning.

## 2 BACKGROUND

### 2.1 MASKED DIFFUSION LANGUAGE MODELS

Diffusion language models (DLMs) are characterized by two processes. The first is a pre-defined corruption (also know as forward) process $q$. This process adds noise to a 'clean' token $\mathbf{x}$ drawn from the data distribution to produce progressively noisy latents $\mathbf{x}_t$, for $t \in [0, 1]$, with noise levels increasing in $t$, and terminates at the fully corrupted latent $\mathbf{x}_1$ drawn from a simple prior. The second process is the learned denoising (backward) process $p_\theta$, which is trained to undo the corruptions from the forward trajectory. Recent discrete diffusion models have focused on forward processes that interpolate between signal and noise (Austin et al., 2021), and in particular they rely on a specific corruption process known as absorbing state / masked diffusion (referred to as MDLMs henceforth) (Sahoo et al., 2024; Shi et al., 2024; Ou et al., 2024). The marginals of this process reflect probability mass iteratively moving away from the data and towards a special mask token $\mathbf{m}$: $q(\mathbf{x}_t \mid \mathbf{x}) = \alpha_t \mathbf{x} + (1 - \alpha_t)\mathbf{m}$, where $\alpha_t$ is a noise schedule monotonically decreasing in $t$. Importantly, both the forward and backward processes are assumed to apply independently along the dimensions of the data, i.e., the tokens in a sequence.

Hoogeboom et al. (2021) and Ou et al. (2024) demonstrate that the MDLM training objective can be transferred into an any-order autoregressive variant, thus enabling the model to generate sequences autoregressively but not necessarily in a left-to-right dependency. Unlike vanilla MDLMs, which conduct fully bidirectional attention, when performing any-order decoding, DLMs output their token probability predictions only conditioned on the tokens that are decoded no later than themselves.

### 2.2 REINFORCEMENT LEARNING WITH VERIFIABLE REWARDS

Policy gradient methods (Williams, 1992; Sutton et al., 1999) have become a central paradigm for improving the reasoning ability of large language models during post-training (Ouyang et al., 2022; Ahmadian et al., 2024; Bai et al., 2022; Li et al., 2023). Starting from a pre-trained model $\pi_{\text{ref}}$, reinforcement learning-based reasoning algorithms optimize a new policy network $\pi_\theta$ by ascending the gradient of the expected reward $r$ for completions generated from $\pi_\theta$ conditioned on an input question $\mathbf{q}$:

$$\nabla_\theta \mathbb{E}_{\mathbf{x}^{1:L} \sim \pi_\theta(\mathbf{x}^{1:L} | \mathbf{q})} \big[ r(\mathbf{x}^{1:L}, \mathbf{q}) \big], \tag{1}$$

where $\mathbf{x}^{1:L}$ denotes a model-generated answer consisting of $L$ tokens. A widely used approach in this context is Group Relative Policy Optimization (GRPO) (Shao et al., 2024), which provides a computationally efficient and low-variance estimator for policy updates. For each query $\mathbf{q}$, GRPO

samples a group of $G$ candidate answers $\{\mathbf{x}_{(1)}^{1:L_{(1)}}, \mathbf{x}_{(2)}^{1:L_{(2)}}, \ldots, \mathbf{x}_{(G)}^{1:L_{(G)}}\}$ from a stale policy $\pi_{\text{old}}$. Each answer is then assigned an advantage value defined as $A_{(i)} = \frac{r(\mathbf{x}_{(i)}^{1:L_{(i)}}, \mathbf{q}) - \text{mean}\{r(\mathbf{x}_{(j)}^{1:L_{(j)}})\}_{1 \leq j \leq G}}{\text{std}\{r(\mathbf{x}_{(j)}^{1:L_{(j)}})\}_{1 \leq j \leq G}}$. GRPO employs the following clipped objective to optimize AR language models:

$$-\mathbb{E}_{\mathbf{x}^{1:L} \sim \pi_{\text{old}}(\cdot | \mathbf{q})} \Big[ \frac{1}{L} \sum_{l=1}^{L} \min \Big\{ \rho^l A^l, \, \text{clip}(\rho^l, 1-\epsilon, 1+\epsilon) A^l \Big\} + \beta D_{\text{KL}} \Big( \pi_\theta(\mathbf{x}^{1:L} \mid \mathbf{q}) \,\|\, \pi_{\text{ref}}(\mathbf{x}^{1:L} \mid \mathbf{q}) \Big) \Big], \quad (2)$$

where $\rho^l = \frac{\pi_\theta(\mathbf{x}^l | \mathbf{x}^{<l}, \mathbf{q})}{\pi_{\text{old}}(\mathbf{x}^l | \mathbf{x}^{<l}, \mathbf{q})}$ denotes the per-token importance ratio, and the same advantage value is assigned to each token in a sequence. The clipping parameter $\epsilon$ constrains policy updates within a trust region, and the KL regularization term penalizes divergence from the reference policy $\pi_{\text{ref}}$. Importantly, due to the causal structure of autoregressive attention masks, the summation over $L$ tokens can be efficiently computed in a single model forward pass.

## 3 METHOD

### 3.1 REINFORCEMENT LEARNING OBJECTIVE FOR DLMS

In contrast to autoregressive (AR) models, whose likelihood accurately factorizes across token positions, the exact likelihood of diffusion language models (DLMs) is computationally intractable. This structural difference renders it theoretically unjustified to directly apply the AR policy gradient formula to DLMs, and makes the derivation of a policy gradient for DLMs a nontrivial problem.

In this section, we introduce our derivation from the policy gradient formula to a modern GRPO-style RL objective for masked DLMs. To begin, we define DLMs' policy gradient objective with respect to the final denoised tokens $\mathbf{x}_0^{1:L}$:

$$\nabla_\theta \mathcal{J}(\theta) = \nabla_\theta \mathbb{E}_{\mathbf{x}_0^{1:L} \sim \pi_\theta(\mathbf{x}_0^{1:L} | \mathbf{q})} \big[ r(\mathbf{x}_0^{1:L}, \mathbf{q}) \big]. \quad (3)$$

Inspired by Black et al. (2023), we marginalize the likelihood over time latents. Moreover, we introduce importance sampling (Kakade & Langford, 2002) to allow reusing of trajectories generated by a stale policy $\pi_{\text{old}}$, which is widely adopted given the computational cost of on-policy sampling. Based on these tricks, we state the following theorem to further simplify $\nabla_\theta \mathcal{J}(\theta)$ (see detailed proof in Appendix 3.1).

**Theorem 3.1.** *At $\theta = \theta_{old}$, $\nabla_\theta \mathcal{J}(\theta)$ admits the following decomposition over latent diffusion steps:*

$$\nabla_\theta \, \mathbb{E}_{\mathbf{x}_{0:T}^{1:L} \sim \pi_{\text{old}}(\mathbf{x}_{0:T}^{1:L} | \mathbf{q})} \left[ r(\mathbf{x}_0^{1:L}, \mathbf{q}) \sum_{t=0}^{T-1} \sum_{l=1}^{L} \mathbf{1}_{\{\mathbf{x}_{t+1}^l = \mathbf{m}, \, \mathbf{x}_t^l \neq \mathbf{m}\}} \frac{\pi_\theta(\mathbf{x}_t^l \mid \mathbf{x}_{t+1}^{1:L}, \mathbf{q})}{\pi_{\text{old}}(\mathbf{x}_t^l \mid \mathbf{x}_{t+1}^{1:L}, \mathbf{q})} \right]. \quad (4)$$

**Remark 3.2.** *In practice, sequences are sampled once from $\pi_{\text{old}}$ and reused for multiple gradient updates. After the first gradient update, the equivalence in Theorem 3.1 is no longer valid and is just an approximation of the true policy gradient. However, this approximation is justified as long as $\theta$ remains close to $\theta_{\text{old}}$, which is typically enforced by restricting the size of each policy update.*

To stabilize training, we adapt Eq. (4) by replacing rewards with advantages (Williams, 1992; Sutton et al., 1999), introducing a clipped trust region (Schulman et al., 2015), and adding a regularization term penalizing divergence from a reference policy (Schulman et al., 2017). Following Zhao et al. (2025), we remove the standard deviation denominator in the advantage computation. In addition, similar to how GRPO averages over sequence lengths, we add a $\frac{1}{L}$ regularization term to cancel out the impact of different sequence lengths within a group. Overall, these operations lead to the GRPO objective for diffusion language models.

**Corollary 3.3.** *The GRPO objective for MDLMs is given by*

$$- \mathbb{E}_{\mathbf{x}_{0:T}^{1:L} \sim \pi_{\text{old}}(\mathbf{x}_{0:T}^{1:L} | \mathbf{q})} \left[ \sum_{t=0}^{T-1} \frac{1}{L} \sum_{l=1}^{L} \mathbf{1}_{\{\mathbf{x}_{t+1}^l = \mathbf{m}, \, \mathbf{x}_t^l \neq \mathbf{m}\}} \min\Big\{ \frac{\pi_\theta(\mathbf{x}_t^l \mid \mathbf{x}_{t+1}^{1:L}, \mathbf{q})}{\pi_{\text{old}}(\mathbf{x}_t^l \mid \mathbf{x}_{t+1}^{1:L}, \mathbf{q})} A^l, \right.$$

$$\left. \text{clip}\big( \frac{\pi_\theta(\mathbf{x}_t^l \mid \mathbf{x}_{t+1}^{1:L}, \mathbf{q})}{\pi_{\text{old}}(\mathbf{x}_t^l \mid \mathbf{x}_{t+1}^{1:L}, \mathbf{q})}, 1-\epsilon, 1+\epsilon \big) A^l \Big\} + \beta D_{\text{KL}} \big( \pi_\theta(\mathbf{x}_{0:T}^{1:L} \mid \mathbf{q}) \,\|\, \pi_{\text{ref}}(\mathbf{x}_{0:T}^{1:L} \mid \mathbf{q}) \big) \right]. \quad (5)$$

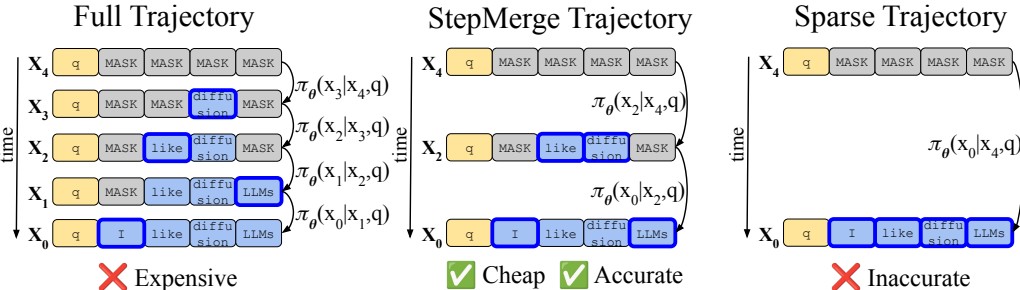

Figure 2: Illustration of our proposed StepMerge strategy. In d2-StepMerge, we cut the trajectory evenly into $N$ time segments and evaluate the likelihood for each segment together. Newly decoded tokens on which we compute the likelihood at the corresponding model forward pass are highlighted.

## 3.2 ESTIMATING THE POLICY GRADIENT FOR DLMS

Unlike AR, DLMs do not employ a left-to-right causal attention mask. Consequently, evaluating the summation over the $T$ diffusion steps in Eq. (5) with a single forward pass is not trivial. For a fully bidirectional DLM, because each step in time is a full sequence $\mathbf{x}_t$, we have to run a full forward pass on $L$ tokens at each time $t$, significantly increasing the cost for computing $\pi(\mathbf{x}_t | \mathbf{x}_{t+1})$. To address this computational bottleneck, we introduce two novel policy gradient estimators tailored for DLMs.

Our first estimator, **d2-StepMerge**, partitions a trajectory into contiguous time segments and evaluates likelihood at specific individual steps. As shown in Section 5.1, the resulting method achieves high performance with a significantly reduced compute budget compared to a full likelihood estimate.

Our second estimator, **d2-AnyOrder**, enables an unbiased likelihood evaluation of a trajectory with a single model forward pass, contingent on the model maintaining an any-order decoding property. Our finding highlights the theoretical and practical significance of any-order causality in advancing diffusion-based reasoning models.

In summary, d2-StepMerge is broadly applicable to MDLMs and offers computational savings with minimal loss of fidelity, whereas d2-AnyOrder is more accurate and efficient, but requires the any-order causality property. In Section 5.2, we provide empirical evidence illustrating the trade-off between these two estimators.

### 3.2.1 ESTIMATOR 1: D2-STEPMERGE

We now present d2-StepMerge. As illustrated in Figure 1, we cut the sampling trajectory of $T$ tokens evenly into $N$ time segments, and optimize the diffusion policy with the following objective:

$$-\mathbb{E}_{\mathbf{x}_{0:T}^{1:L} \sim \pi_{\text{old}}(\mathbf{x}_{0:T}^{1:L}|\mathbf{q})} \left[ \sum_{n=0}^{N-1} \frac{1}{L} \sum_{l=1}^{L} \mathbf{1}_{\{\mathbf{x}_{(n+1)\frac{L}{N}}^l = \mathbf{m}, \mathbf{x}_{n\frac{L}{N}}^l \neq \mathbf{m}\}} \min\{ \frac{\pi_\theta(\mathbf{x}_{n\frac{L}{N}}^l \mid \mathbf{x}_{(n+1)\frac{L}{N}}^{1:L}, \mathbf{q})}{\pi_{\text{old}}(\mathbf{x}_{n\frac{L}{N}}^l \mid \mathbf{x}_{(n+1)\frac{L}{N}}^{1:L}, \mathbf{q})} A^l, \right.$$

$$\left. \text{clip}(\frac{\pi_\theta(\mathbf{x}_{n\frac{L}{N}}^l \mid \mathbf{x}_{(n+1)\frac{L}{N}}^{1:L}, \mathbf{q})}{\pi_{\text{old}}(\mathbf{x}_{n\frac{L}{N}}^l \mid \mathbf{x}_{(n+1)\frac{L}{N}}^{1:L}, \mathbf{q})}, 1 - \epsilon, 1 + \epsilon) A^l \} + \beta D_{\text{KL}}\big(\pi_\theta(\mathbf{x}_{0:T}^{1:L} \mid \mathbf{q}) \,\|\, \pi_{\text{ref}}(\mathbf{x}_{0:T}^{1:L} \mid \mathbf{q})\big) \right]. \quad (6)$$

**Understanding the role of $N$.** With d2-StepMerge, we reduce the number of model passes from $T$ to $N$. A natural question to ask is what is the effect of $N$ on the accuracy of the policy gradient? Since the importance weight is the quotient of two trajectory likelihood evaluations, we further narrow this down to studying the discrepancy between the complete likelihood decomposition of the trajectory and that yielded by our StepMerge strategy. Formally, we define the KL divergence between the complete and StepMerge decompositions by a masked DLM policy $\pi_\theta$ as $D_N$:

$$D_N(\pi_\theta) = D_{\text{KL}}\big(\prod_{t=0}^{T-1} \prod_{\substack{l \text{ s.t.} \\ \mathbf{x}_{t+1}^l = \mathbf{m}, \mathbf{x}_t^l \neq \mathbf{m}}} \pi_\theta(\mathbf{x}_t^l \mid \mathbf{x}_{t+1}^{1:L}, \mathbf{q}) \| \prod_{n=0}^{N-1} \prod_{\substack{l \text{ s.t.} \\ \mathbf{x}_{(n+1)\frac{L}{N}}^l = \mathbf{m}, \mathbf{x}_{n\frac{L}{N}}^l \neq \mathbf{m}}} \pi_\theta(\mathbf{x}_{n\frac{L}{N}}^l \mid \mathbf{x}_{(n+1)\frac{L}{N}}^{1:L}, \mathbf{q})\big).$$

$$(7)$$

We compute $D_N$ for LLaDA-8B-Instruct (Nie et al., 2025) on the test sets of four datasets (introduced in Section 5.1). As shown in Figure 3, $D_N$ decreases monotonically with increasing $N$, indicating that d2-StepMerge trades compute for likelihood estimation precision.

**Remark 3.4.** *diffu-GRPO (Zhao et al., 2025) is a special case of d2-StepMerge where $N = 1$. As noted above, $N = 1$ produces an inaccurate likelihood estimation and may thus harm the performance of RL. This is consistent with our experimental results shown in Section 5.1.*

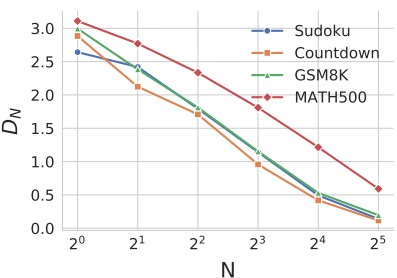

Figure 3: $D_N(\pi_{\text{LLaDA}})$ for varying $N$.

### 3.2.2 ESTIMATOR 2: D2-ANYORDER

In this section, we present d2-AnyOrder, our second policy gradient estimator that can obtain likelihood estimation of a trajectory losslessly and with one model pass. Importantly, d2-AnyOrder is contingent on an any-order decoding property of MDLMs. Therefore, we first briefly recap any-order decoding, based on which we propose our one-shot likelihood evaluation. We then formally introduce the d2-AnyOrder objective. Moreover, we examine the application scope of d2-AnyOrder and discover that although previous works (Hoogeboom et al., 2021; Ou et al., 2024) have implied an equivalence between fully bidirectional MDLMs and any-order MDLMs, applying any-order decoding to the former type of models is nontrivial in practice. This observation encourages us to provide a more detailed investigation of the any-order causality feature of MDLMs (see Section 4.2).

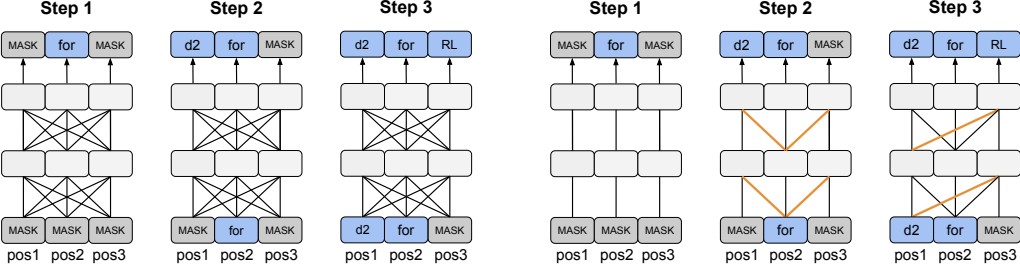

(a) Fully bidirectional MDLM decoding.  (b) Any-order decoding.

Figure 4: Illustration of different DLM decoding strategies. We depict attention with query tokens (one layer up) attending to keys/values (one layer below) via an undirected connected line. The output at each position is depicted with a directed arrow. "pos" refers to positional encoding index. We use a three token example where the decoding order is "for→d2→RL". At each time step, newly added attention relations in any-order decoding are highlighted with red line markers.

**Notations.** Given a token trajectory $\mathbf{x}_{0:T}^{1:L}$, for each position $l$, we use $t_l$ to denote the time step when $\mathbf{m}^l$ is decoded as $\mathbf{x}_0^l$. We define $\Omega_l = \{l' \mid t_{l'} > t_l\}$. Intuitively, $\Omega_l$ includes all the positions decoded before $\mathbf{x}_0^l$ because $t$ goes from $T$ to $0$ during sampling. We then denote $\{\mathbf{x}_0^l \mid l \in \Omega_l\}$ as $\mathbf{x}_0^{\Omega_l}$. Token positions decoded at time step $t$ are denoted as $U_t$ and $\mathbf{x}_0^{U_t} = \{\mathbf{x}_0^l \mid l \in U_t\}$.

**Any-order decoding.** In the vanila MDLM decoding process (Figure 4a), all tokens can attend to each other, making the attention pattern fully bidirectional. Extending Hoogeboom et al. (2021), we propose a different decoding strategy termed any-order decoding (Figure 4b). Specifically, a mask token $\mathbf{m}^l$ attends to $\mathbf{x}_0^{\Omega_l} \cup \{\mathbf{m}^l\}$ when it is decoded. After it decodes to

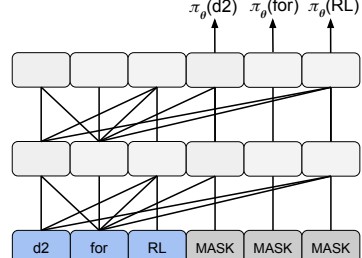

Figure 5: Illustration of one-shot trajectory likelihood evaluation. Continuation of the example from Figure 4.

$\mathbf{x}_0^l$, it attends to $\mathbf{x}_0^{\Omega_l} \cup \mathbf{x}^{U_{t_l}}$ (the set of tokens unmasked previously and concurrently to that token), and this attention pattern stays unchanged for the rest of the sampling process. Note that our decoding method allows multiple tokens to be decoded every time step (see Appendix B.1 for more details).

**One-shot trajectory likelihood evaluation.** Recall that the likelihood of a trajectory $\mathbf{x}_{0:T}^{1:L}$ generated via any-order decoding is $\pi_\theta(\mathbf{x}_{0:T}^{1:L}) = \prod_{t=0}^{T-1} \prod_{l \in U_t} \pi_\theta(\mathbf{x}_t^l \mid \mathbf{x}_{t+1}^{1:L})$. We propose that the above likelihood can be computed in one forward pass. To efficiently implement this computation, we construct a $2L$-length sequence $\mathbf{x}_0^{1:L} \oplus \mathbf{m}^{L+1:2L}$, where $\oplus$ denotes concatenation along the sequence dimension and $\mathbf{m}^{L+1:2L}$ are mask tokens. The positional encodings are assigned as $\text{pos}_l = l \bmod L$ for $1 \le l \le 2L$, so that each token–mask pair shares the same position index. We then define the attention mask so that as a query token, $\mathbf{x}_l$ attends to $\mathbf{x}^{\Omega_l} \cup \mathbf{x}^{U_{t_l}}$, and $\mathbf{m}^{L+l}$ attends to $\mathbf{x}^{\Omega_l} \cup \{\mathbf{m}^{L+l}\}$ As illustrated in Figure 5, the output logits of the mask token at position $L + l$ reproduce exactly the logits of $\mathbf{x}_0^l$ as if it had been decoded during sampling. We therefore denote the resulting likelihood estimate as $\pi_\theta^{\text{AO}}(\mathbf{x}_0^l \mid \mathbf{x}_0^{1:L} \oplus \mathbf{m}^{L+1:2L})$. More details are provided in Appendix B.3.

**d2-AnyOrder.** We hereby introduce d2-AnyOrder. For a DLM that applies to any-order decoding, we propose to train the policy network with the following d2-AnyOrder objective:

$$-\mathbb{E}_{\mathbf{x}_{0:T}^{1:L} \sim \pi_{\text{old}}(\mathbf{x}_{0:T}^{1:L} \mid \mathbf{q})} \left[ \frac{1}{L} \sum_{l=1}^{L} \min\left\{ \frac{\pi_\theta^{\text{AO}}(\mathbf{x}_0^l \mid \mathbf{x}_0^{1:L} \oplus \mathbf{m}^{L+1:2L}, \mathbf{q})}{\pi_{\text{old}}^{\text{AO}}(\mathbf{x}_0^l \mid \mathbf{x}_0^{1:L} \oplus \mathbf{m}^{L+1:2L}, \mathbf{q})} A^l, \right. \right.$$

$$\left. \left. \text{clip}\left( \frac{\pi_\theta^{\text{AO}}(\mathbf{x}_0^l \mid \mathbf{x}_0^{1:L} \oplus \mathbf{m}^{L+1:2L}, \mathbf{q})}{\pi_{\text{old}}^{\text{AO}}(\mathbf{x}_0^l \mid \mathbf{x}_0^{1:L} \oplus \mathbf{m}^{L+1:2L}, \mathbf{q})}, 1 - \epsilon, 1 + \epsilon \right) A^l \right\} + \beta\, D_{\text{KL}}\left( \pi_\theta(\mathbf{x}_{0:T}^{1:L} \mid \mathbf{q}) \,\|\, \pi_{\text{ref}}(\mathbf{x}_{0:T}^{1:L} \mid \mathbf{q}) \right) \right].$$

$$(8)$$

**Application Scope of d2-AnyOrder.** Since d2-AnyOrder relies on any-order decoding, which is not the default sampler for many DLMs, it is necessary to explore the types of DLMs to which d2-AnyOrder applies. Previous works (Hoogeboom et al., 2021; Sahoo et al., 2025b) have proposed special training algorithms for any-order DLMs. Thus, d2-AnyOrder naturally applies to DLMs trained with these algorithms. Although Hoogeboom et al. (2021); Ou et al. (2024) have shown that the training objective of any-order DLMs is equivalent to that of fully bidirectional MDLMs, we discover empirically that, in practice, any-order decoding does not apply to models trained with the latter objective, such as LLaDA.

Table 1: Average per token log-likelihood for sequences generated by $\pi_{\text{ref}}$ and evaluated under $\pi_{\text{ref}}$ and the one-shot any-order version of the same model $\pi_{\text{ao}}$.

| | Per token LL. | $D_{\text{KL}}(\pi_{\text{ref}}\|\pi_{\text{ao}})$ |
|---|---|---|
| $\pi_{\text{ref}}$ | -0.128 | — |
| $\pi_{\text{ao}}$ | -3.051 | 2.334 |

We first apply any-order decoding to LLaDA directly and observe that LLaDA generates the EOS token immediately when the decoding position does not attend to any mask tokens to its right. In addition, we design another experiment where we generate completions to questions from the GSM8K test set using LLaDA-8B-Instruct, and then compute the trajectory log-likelihood (LL.) by the full trajectory decomposition (denoted as $\pi_{\text{ref}}$) as ground-truth, and the one-shot trick (denoted as $\pi_{\text{ao}}$). We also compute the $D_{\text{KL}}$ between $\pi_{\text{ref}}$ and $\pi_{\text{ao}}$.

In Table 1, we see that the LL. estimate attained from the one-shot trick is an order of magnitude different than that obtained from the ground-truth $\pi_{\text{ref}}$. Hence, it is not practical to directly apply the one-shot any-order trick to LLaDA. We hypothesize that this is because LLaDA allows tokens to attend to mask token positions decoded after them. This attention design, although not necessary from a probabilistic perspective, has enabled the neural network to store important information in the hidden states of mask tokens, thereby limiting LLaDA's potential for RL post-training.

## 4 THEORETICAL ANALYSIS

### 4.1 D2-STEPMERGE

In this section, we introduce a bound for $D_N$ (defined in Section 3.2.1) to quantify the compute-bias tradeoff of d2-StepMerge. Our upper bound monotonically decreases as $N$ increases, theoretically justifying our observation in Figure 3.

**Theorem 4.1** (Approximation Error Bound). *Suppose $\pi_\theta$ has a fixed schedule where $L$ tokens are unmasked over $T$ timesteps. The KL divergence $D_N(\pi_\theta)$ between the full trajectory and StepMerge approximation is bounded by:*

$$D_N(\pi_\theta) \le L \cdot \log\left(\frac{T}{N} + 1\right) + L \cdot \epsilon_{block} \tag{9}$$

*where $\epsilon_{block}$ is a constant measuring how much the model's token predictions can change when we skip intermediate diffusion steps within a block (see our proof in Appendix A.2).*

### 4.2 D2-ANYORDER

In this section, we provide a more detailed definition of the any-order causality feature of DLMs. Assuming a Transformer architecture, we denote the token positions that query token $\mathbf{x}_t^l$ attends to at time step $t$ as $A_t^l$. Additionally, we assume that when decoding a token, a DLM policy should attend to all previously unmasked tokens to maximize information utilization. Formally, inheriting the notation system from Section 3.2.2, we define a DLM as *reasonable* if $\forall 1 \leq l \leq L, \forall 0 \leq t \leq t_l, \Omega_l \subseteq A_t^l$.

**Definition 4.2.** A *reasonable* Transformer DLM policy $\pi_\theta$ is *any-order causal* if when decoding a trajectory $\mathbf{x}_{0:T}^{1:L}$, for all $1 \leq l \leq L$ the attention pattern satisifies:

1. **Unmasked tokens' attentions are consistent.** $\forall 0 \leq t_i \neq t_j < t_l$, we have $A_{t_i}^l = A_{t_j}^l$,
2. **Tokens don't attend to future when decoded.** $A_{t_l}^l \subseteq \Omega_l \cup \{l\}$,
3. **Unmasked tokens don't attend to future.** if $\mathbf{x}^l$ is not within the last two decoding steps, then $\forall 0 \leq t < t_l, A_t^l \subseteq \Omega_l \cup U_{t_l}$.

**Theorem 4.3.** *For any Transformer networks with more than one layer, a DLM trajectory likelihood can be computed by one model pass iff. $\pi_\theta$ is any-order causal (proof provided in Appendix A.3).*

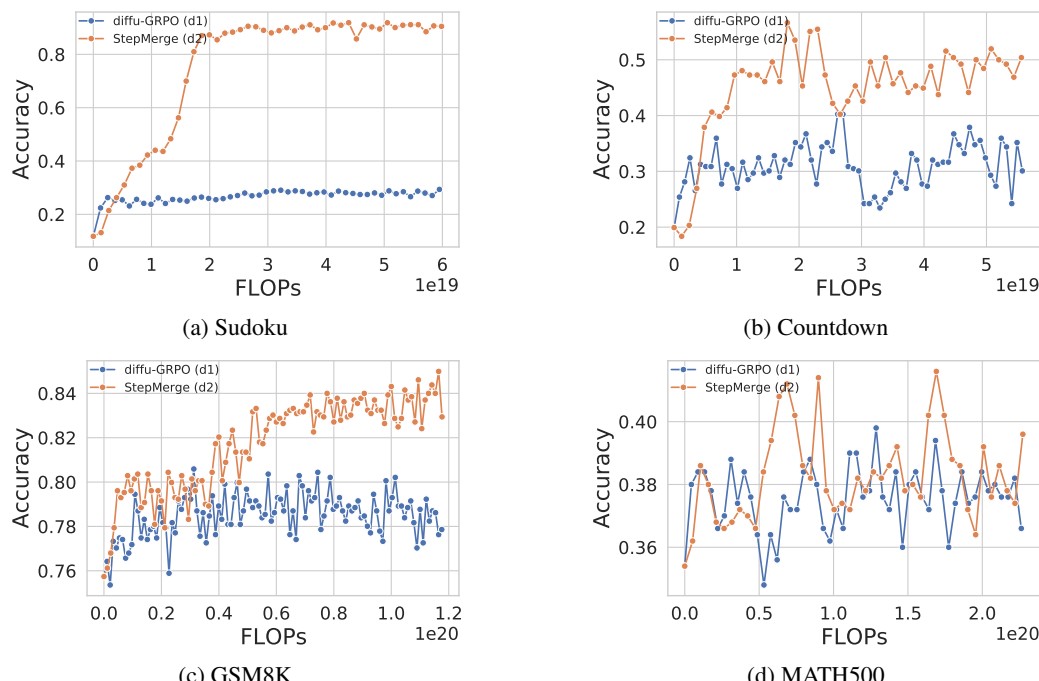

Figure 6: Performance-compute dynamics of d2-StepMerge on four reasoning benchmarks.

## 5 EXPERIMENTS

### 5.1 D2-STEPMERGE

**Experimental Setup.** To test d2-StepMerge's effect on improving DLM's reasoning capacity, we follow Zhao et al. (2025) in using LLaDA-8B-Instruct (Nie et al., 2025) as the base model and apply different RL post-training algorithms on top of it for four reasoning benchmarks, including two mathematical reasoning benchmarks (GSM8K (Cobbe et al., 2021) and MATH500 (Lightman et al., 2023)), and two logical reasoning benchmarks (Countdown and Sudoku). For both training and evaluation, two tokens are generated at each time step. We use a group size of 6, and each batch contains 16 questions. We track the compute requirements (measured in Floating-Point Operations, i.e., FLOPs) and evaluate the test set performance of different methods for every fixed FLOP interval.

**Baselines.** To benchmark our proposed framework, we compare d2 against a diverse set of baselines. Specifically, we include the original LLaDA, LLaDA 1.5 (Zhu et al., 2025), a low-variance direct preference optimization (Rafailov et al., 2023) post-training algorithm, d1 (Zhao et al., 2025), a hybrid method that combines supervised fine-tuning on the s1k (Muennighoff et al., 2025) long chain-of-thought data with diffu-GRPO, as well as wd1 (Tang et al., 2025), which reformulates policy optimization as a weighted likelihood objective to eliminate the dependence on policy ratios.

**Results.** As shown in Figure 6, d2-StepMerge consistently outperforms diffu-GRPO in four reasoning benchmarks. In Sudoku, Countdown, and GSM8K, d2-StepMerge significantly dominates diffu-GRPO, and in MATH500, d2-StepMerge demonstrates a better trend. These results indicate that d2 achieves a superior trade-off between efficiency and performance. Moreover, Table 2 shows that even without supervised fine-

Table 2: Benchmark performance of different reasoning frameworks. † indicates results evaluated on the released checkpoint. ‡ indicates results taken from the corresponding paper. Baselines that include post-training are shaded. Best results are **bolded**.

| Method | Sudoku | Countdown | GSM8K | MATH500 |
|---|---|---|---|---|
| LLaDA[†] | 11.8% | 19.9% | 75.7% | 35.4% |
| LLaDA 1.5[†] | 12.5% | 23.4% | 78.6% | 36.8% |
| d1[‡] | 22.1% | 42.2% | 82.1% | 40.2% |
| wd1[‡] | 25.2% | 51.2% | 82.3% | 39.0% |
| d2 (ours) | **91.9%** | **56.6%** | **85.0%** | **41.6%** |

tuning on extra chain-of-thought data, d2 can outperform existing diffusion reasoning frameworks, demonstrating the efficacy of our proposed likelihood evaluation strategy.

**Ablation: Varying $N$ in d2-StepMerge.** In Figure 7, we show the Accuracy-FLOP trade-off of d2-StepMerge with different values of $N$ in the Sudoku benchmark. With a small $N$, the model does not reliably converge to competitive performance because the corresponding likelihood estimation is highly inaccurate. In contrast, excessively large $N$ leads to over-allocation of compute to likelihood estimation, resulting in slower convergence. Our results identify $N = 16$ as a favorable balance: it achieves performance comparable to $N = 32$ and $N = 64$, while requiring substantially fewer FLOPs.

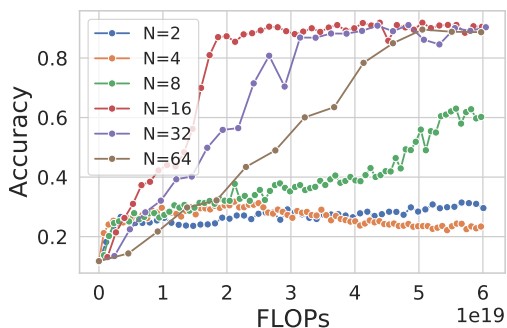

Figure 7: d2-StepMerge performance in Sudoku with different $N$.

## 5.2 D2-ANYORDER

**Experimental Setup.** We take Eso-LM (Sahoo et al., 2025b), a special type of DLMs whose any-order causal property is enforced by its training algorithms, and test d2-AnyOrder on top of it. We use a 190M parameter model trained on OpenWebText (Gokaslan et al., 2019). Following Singhal et al. (2025), we test d2-AnyOrder's capacity to steer the model towards toxic output, which is rare in the model's normal behavior and useful in the red-teaming scenario. We utilize an existing toxicity classifier (Logacheva et al., 2022) as the reward model, and the RL algorithm should increase the toxicity score. In both training and evaluation, we let the model generate sentences with 512 tokens in free form. We set the group size as 16, and each group consists of a batch of 4 samples. Similarly to Section 5.1, we set the compute budget (measured by FLOPs) constant and evaluate the model every fixed FLOP interval. For each checkpoint, we let the model generate 512 new token sequences, and evaluate their average toxicity score under the reward model and their generative perplexity (Gen PPL. ↓) under GPT2-XL (Radford et al., 2019).

**Baselines.** In this setting, generation is in free form without prompts; the likelihood evaluation in diffu-GRPO and wd1 cannot provide helpful information. Thus, following Su et al. (2025), we select DDPO (Black et al., 2023), a PPO-style algorithm that also decomposes the diffusion policy gradient along $t$, as our baseline. Since d2 is a GRPO-style algorithm, we remove the value network in DDPO and use the group advantage instead. In addition, DDPO does not involve trust regions or KL divergence with the reference model, so we also remove the corresponding parts in d2 to conduct a fair comparison.

**Results.** As shown in Figure 8, under the same compute budget, d2 substantially outperforms DDPO in toxicity score, reaching near-zero values while DDPO remains below –8. Though the GPT2-XL generative perplexity of d2 decreases more than that of DDPO—suggesting greater divergence from the reference model—we regard this as reasonable given the substantially stronger steering applied to d2.

**Ablation: Comparing d2-AnyOrder and d2-StepMerge.** Figure 9 compares the two d2 variants on toxicity steering. For d2-StepMerge, we fix $N = 8$, as it performs well in earlier experiments. d2-AnyOrder achieves a slightly better toxicity score trend. Besides, d2-AnyOrder shows smaller drops in Gen PPL. at comparable toxicity levels, indicating greater stability which is largely due to its unbiased estimation. In practice, when the DLM satisfies the any-order causality property, d2-AnyOrder is preferable; otherwise, d2-StepMerge provides a more general solution.

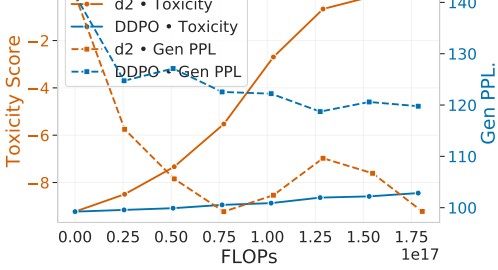

Figure 8: Toxicity steering results of d2-AnyOrder.

# 6 RELATED WORK, DISCUSSION, AND CONCLUSION

**RL for DLMs.** Diffusion language models (Sahoo et al., 2024; 2025a; Nie et al., 2025; Ye et al., 2025) have recently emerged as a strong alternative to AR, motivating a line of works on reinforcement learning for DLMs. A central challenge is to accurately estimate token likelihood. diffu-GRPO (Zhao et al., 2025) approximates the likelihood using logits from an input of all mask tokens. DiffuCoder's coupled-GRPO (Gong et al., 2025) proposes to improve this estimate by randomly splitting sequences into two segments. They also incorporate the $\frac{1}{t}$ weighting factor from MDLM's ELBO (Sahoo et al., 2024) into their RL objective. wd1 (Tang

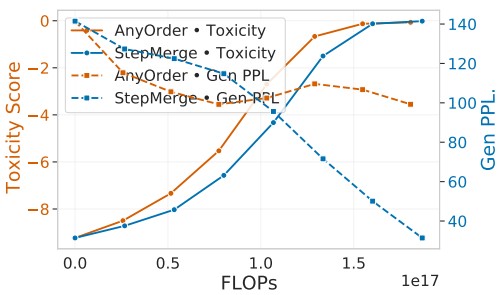

Figure 9: Comparison of d2-AnyOrder and d2-StepMerge on toxicity steering.

et al., 2025) reformulates policy optimization as a weighted likelihood objective and inherits the likelihood estimation method of diffu-GRPO. LLaDOU (Huang et al., 2025) decomposes the likelihood along the diffusion steps, though the direct approach is computationally prohibitive. Concurrent to our work, TraceRL (Wang et al., 2025b) proposes merging timesteps for trajectory likelihood evaluation. While TraceRL and our d2-StepMerge share the same objective, our method further justifies this approach by establishing bounds on the KL divergence.

**Limitations and Future Work.** While d2-StepMerge achieves a favorable performance–efficiency trade-off among diffusion language models, it still requires multiple forward passes and yields a biased estimation, making it less optimal than autoregressive approaches. In contrast, d2-AnyOrder is restricted to DLMs trained under specific constraints, limiting its general applicability. An important next step is to investigate how to enforce the any-order causality property on pretrained checkpoints such as LLaDA. Moreover, although our experiments show that the d2 RL algorithm alone can significantly enhance reasoning, it remains an open question whether large-scale post-training on DLMs requires additional supervised finetuning, or whether an RL-only approach, akin to DeepSeek-R1-Zero (Guo et al., 2025), can be effectively applied in this setting.

**Conclusion.** In this work, we introduce d2, a principled RL framework for diffusion language models, derived directly from the policy gradient formula with a transparent mathematical foundation. To enable trajectory likelihood computation in the RL objective, we propose two customized estimators and provide supporting theoretical analysis. We further explore the under-examined property of any-order causality in DLMs, demonstrating its significance for enhancing reasoning capabilities. Empirically, d2 achieves state-of-the-art performance on reasoning benchmarks for DLMs under matched FLOP budgets, without relying on supervised chain-of-thought finetuning.

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

CONTENTS

## A   THEORETICAL RESULTS

### A.1   PROOF OF THEOREM 3.1

*Proof.* We begin by recalling the definition of the RL objective:

$$\mathcal{J}(\theta) = \mathbb{E}_{\mathbf{x}_0^{1:L} \sim \pi_\theta(\mathbf{x}_0^{1:L}|\mathbf{q})}\left[r(\mathbf{x}_0^{1:L}, \mathbf{q})\right]. \tag{10}$$

Taking the policy gradient and expanding the expectation over the trajectory distribution, we have

$$
\nabla_\theta \mathcal{J}(\theta) = \nabla_\theta \sum_{\mathbf{x}_0^{1:L}} \pi_\theta(\mathbf{x}_0^{1:L} \mid \mathbf{q}) \, r(\mathbf{x}_0^{1:L}, \mathbf{q})
$$

$$
= \nabla_\theta \sum_{\mathbf{x}_0^{1:L}} \sum_{\mathbf{x}_{1:T}^{1:L}} \pi_\theta(\mathbf{x}_0^{1:L} \mid \mathbf{x}_{1:T}^{1:L}, \mathbf{q}) \, \pi_\theta(\mathbf{x}_{1:T}^{1:L} \mid \mathbf{q}) \, r(\mathbf{x}_0^{1:L}, \mathbf{q})
$$

$$
= \nabla_\theta \sum_{\mathbf{x}_{0:T}^{1:L}} \pi_\theta(\mathbf{x}_{0:T}^{1:L} \mid \mathbf{q}) \, r(\mathbf{x}_0^{1:L}, \mathbf{q})
$$

$$
= \nabla_\theta \, \mathbb{E}_{\mathbf{x}_{0:T}^{1:L} \sim \pi_\theta(\mathbf{x}_0^{1:L} \mid \mathbf{q})} \left[ r(\mathbf{x}_0^{1:L}, \mathbf{q}) \right]. \tag{11}
$$

Introducing importance sampling with respect to the stale policy $\pi_{\mathrm{old}}$, we obtain

$$
\nabla_\theta \mathcal{J}(\theta) = \nabla_\theta \, \mathbb{E}_{\mathbf{x}_{0:T}^{1:L} \sim \pi_{\mathrm{old}}(\mathbf{x}_{0:T}^{1:L} \mid \mathbf{q})} \left[ r(\mathbf{x}_0^{1:L}, \mathbf{q}) \, \frac{\pi_\theta(\mathbf{x}_{0:T}^{1:L} \mid \mathbf{q})}{\pi_{\mathrm{old}}(\mathbf{x}_{0:T}^{1:L} \mid \mathbf{q})} \right]. \tag{12}
$$

By exploiting the Markov factorization of the backward process, the joint distribution decomposes as

$$
\pi_\theta(\mathbf{x}_{0:T}^{1:L} \mid \mathbf{q}) = \pi(\mathbf{x}_T^{1:L}) \prod_{t=0}^{T-1} \pi_\theta(\mathbf{x}_t^{1:L} \mid \mathbf{x}_{t+1}^{1:L}, \mathbf{q}). \tag{13}
$$

Substituting this factorization, we obtain

$$
\nabla_\theta \mathcal{J}(\theta) = \mathbb{E}_{\mathbf{x}_{0:T}^{1:L} \sim \pi_{\mathrm{old}}(\mathbf{x}_{0:T}^{1:L} \mid \mathbf{q})} \left[ r(\mathbf{x}_0^{1:L}, \mathbf{q}) \, \nabla_\theta \frac{\prod_{t=0}^{T-1} \pi_\theta(\mathbf{x}_t^{1:L} \mid \mathbf{x}_{t+1}^{1:L}, \mathbf{q})}{\prod_{t=0}^{T-1} \pi_{\mathrm{old}}(\mathbf{x}_t^{1:L} \mid \mathbf{x}_{t+1}^{1:L}, \mathbf{q})} \right]. \tag{14}
$$

In MDLM, we assume independence across token positions, the expression further decomposes as

$$
\nabla_\theta \mathcal{J}(\theta) = \mathbb{E}_{\mathbf{x}_{0:T}^{1:L} \sim \pi_{\mathrm{old}}(\mathbf{x}_{0:T}^{1:L} \mid \mathbf{q})} \left[ r(\mathbf{x}_0^{1:L}, \mathbf{q}) \, \nabla_\theta \prod_{t=0}^{T-1} \prod_{l=1}^{L} \frac{\pi_\theta(\mathbf{x}_t^l \mid \mathbf{x}_{t+1}^{1:L}, \mathbf{q})}{\pi_{\mathrm{old}}(\mathbf{x}_t^l \mid \mathbf{x}_{t+1}^{1:L}, \mathbf{q})} \right]. \tag{15}
$$

Expanding the gradient of the product yields

$$
\nabla_\theta \mathcal{J}(\theta) = \mathbb{E}_{\mathbf{x}_{0:T}^{1:L} \sim \pi_{\mathrm{old}}(\mathbf{x}_{0:T}^{1:L} \mid \mathbf{q})} \left[ r(\mathbf{x}_0^{1:L}, \mathbf{q}) \sum_{t=0}^{T-1} \sum_{l=1}^{L} \nabla_\theta \frac{\pi_\theta(\mathbf{x}_t^l \mid \mathbf{x}_{t+1}^{1:L}, \mathbf{q})}{\pi_{\mathrm{old}}(\mathbf{x}_t^l \mid \mathbf{x}_{t+1}^{1:L}, \mathbf{q})} \prod_{\substack{t' \neq t \\ l' \neq l}} \frac{\pi_\theta(\mathbf{x}_{t'}^{l'} \mid \mathbf{x}_{t'+1}^{1:L}, \mathbf{q})}{\pi_{\mathrm{old}}(\mathbf{x}_{t'}^{l'} \mid \mathbf{x}_{t'+1}^{1:L}, \mathbf{q})} \right].
$$

$$
\tag{16}
$$

When evaluated at $\theta = \theta_{\mathrm{old}}$, the product term reduces to 1, yielding

$$
\nabla_\theta \mathcal{J}(\theta) \Big|_{\theta = \theta_{\mathrm{old}}} = \mathbb{E}_{\mathbf{x}_{0:T}^{1:L} \sim \pi_{\mathrm{old}}(\mathbf{x}_{0:T}^{1:L} \mid \mathbf{q})} \left[ r(\mathbf{x}_0^{1:L}, \mathbf{q}) \, \nabla_\theta \sum_{t=0}^{T-1} \sum_{l=1}^{L} \frac{\pi_\theta(\mathbf{x}_t^l \mid \mathbf{x}_{t+1}^{1:L}, \mathbf{q})}{\pi_{\mathrm{old}}(\mathbf{x}_t^l \mid \mathbf{x}_{t+1}^{1:L}, \mathbf{q})} \right]. \tag{17}
$$

Finally, since the model $\pi_\theta$ is only invoked when a masked token $\mathbf{m}$ becomes unmasked, the objective simplifies to

$$
\nabla_\theta \mathcal{J}(\theta) \Big|_{\theta = \theta_{\mathrm{old}}} = \nabla_\theta \, \mathbb{E}_{\mathbf{x}_{0:T}^{1:L} \sim \pi_{\mathrm{old}}(\mathbf{x}_{0:T}^{1:L} \mid \mathbf{q})} \left[ r(\mathbf{x}_0^{1:L}, \mathbf{q}) \sum_{t=0}^{T-1} \sum_{l=1}^{L} \mathbf{1}_{\{\mathbf{x}_{t+1}^l = \mathbf{m}, \, \mathbf{x}_t^l \neq \mathbf{m}\}} \frac{\pi_\theta(\mathbf{x}_t^l \mid \mathbf{x}_{t+1}^{1:L}, \mathbf{q})}{\pi_{\mathrm{old}}(\mathbf{x}_t^l \mid \mathbf{x}_{t+1}^{1:L}, \mathbf{q})} \right].
$$

$$
\tag{18}
$$

$\square$

## A.2 Proof of Theorem 4.1

Our proof proceeds in three key steps:

1. **Decompose the diffusion process**: We factor each diffusion step into timing (which tokens unmask) and value (what values they take) components, exploiting the conditional independence structure.

2. **Bound consecutive timesteps**: For adjacent timesteps, we prove the timing component contributes at most $2k \cdot \log 2$ bits (where $k$ tokens unmask), while the value component vanishes under mild assumptions.

3. **Extend to full trajectory**: We aggregate bounds over $N$ blocks, showing each block contributes at most $k_n \log B$ bits, yielding the final $O(U \log(T/N))$ bound.

To keep notation concise, we define a sequence *without* a superscript as $x_t = x_t^{1:L} = [x_t^1, \ldots x_t^L]$ and drop the prompt $q$ that we condition on $\pi_\theta(\cdot \mid \cdot, q)$.

### A.2.1 TIMING AND VALUE FACTORIZATION

The reverse process, $\pi_\theta(x_t \mid x_{t+1})$, can be decomposed into two conceptual and computational steps: a *timing* decision of whether to unmask a token, followed by a *value* assignment of what it becomes. This allows us to factor the distribution into two simpler components.

**Definition A.1** (Timing and Value Decomposition). We introduce an indicator variable $S_t^l = \mathbf{1}[x_{t+1}^l = \mathbf{m} \wedge x_t^l \neq \mathbf{m}]$ for the unmasking event, and a categorical random variable $V_t^l = x_t^l$ for the token's value. They are aggregated as $S_t = [S_t^1 \ldots S_t^L]$ and $V_t = [V_t^1 \ldots V_t^L]$ and factorize the reverse process:

$$\pi_\theta(x_t \mid x_{t+1}) = \underbrace{\tau(S_t \mid x_{t+1})}_{\text{Timing}} \cdot \underbrace{\nu_\theta(V_t \mid S_t^l, x_{t+1})}_{\text{Value}}. \tag{19}$$

The reverse processes decomposes over tokens as $\pi_\theta(x_t \mid x_{t+1}) = \prod_{l=1}^L \pi_\theta(x_t^l \mid x_{t+1}^{1:L})$ which emits a similar per-token factorization $\pi_\theta(x_t^l \mid x_{t+1}^{1:L}) = \tau(S_t^l \mid x_{t+1}^l) \cdot \nu_\theta(V_t^l \mid S_t^l, x_{t+1}^{1:L})$. We now define $\tau$ and $\nu_\theta$ as follows.

**Timing Distribution ($\tau$):** This distribution models the probability of the unmasking event $S_t^l$ for any unmasking schedule (e.g. greedy, ancestral, top-k, etc.).

- If a token at $t + 1$ is already unmasked ($x_{t+1}^l \neq \mathbf{m}$), it cannot unmask again; thus, the event $S_t^l = 1$ has zero probability.

- If a token is masked ($x_{t+1}^l = \mathbf{m}$), it unmasks with probability $\alpha_t$ determined by the unmasking schedule.

$$\tau(S_t^l = s \mid x_{t+1}^l) = \begin{cases} \alpha_t & \text{if } s = 1 \text{ and } x_{t+1}^l = \mathbf{m} \\ 1 - \alpha_t & \text{if } s = 0 \text{ and } x_{t+1}^l = \mathbf{m} \\ 0 & \text{if } s = 1 \text{ and } x_{t+1}^l \neq \mathbf{m} \\ 1 & \text{if } s = 0 \text{ and } x_{t+1}^l \neq \mathbf{m} \end{cases} \tag{20}$$

**Value Distribution ($\nu_\theta$):** This distribution assigns a value $V_t^l$ to the token, conditional on the timing decision $S_t^l$ and the full sequence context $x_{t+1}$.

- If the decision was to unmask ($S_t^l = 1$), $\nu_\theta$ is the categorical distribution over the vocabulary $\mathcal{V}$ given by the softmax output of the neural network $f_\theta$.

- If the decision was to not unmask ($S_t^l = 0$), the process is deterministic: the token's state from $t + 1$ is simply preserved at time $t$.

$$\nu_\theta(V_t^l = v \mid S_t^l, x_{t+1}) = \begin{cases} \text{softmax}(f_\theta(x_{t+1})_l)_v & \text{if } S_t^l = 1 \\ \delta_{v, x_{t+1}^l} & \text{if } S_t^l = 0 \end{cases} \tag{21}$$

where $\delta_{a,b}$ is the Kronecker delta, enforcing the deterministic state preservation when $S_t^l = 0$.

### A.2.2 CONSECUTIVE TIMESTEP BOUNDS

We begin by analyzing the error over two consecutive timesteps $t$ and $t + 1$. This allows for the simplest possible analysis between the full trajectory and stepmerge trajectory:

$$p_{\text{true}}(x_t, x_{t+1} \mid x_{t+2}) = \pi_\theta(x_t \mid x_{t+1})\pi_\theta(x_{t+1} \mid x_{t+2}) \quad \text{(Full Trajectory)} \tag{22}$$

$$p_{\text{approx}}(x_t, x_{t+1} \mid x_{t+2}) = \pi_\theta(x_t \mid x_{t+2})\pi_\theta(x_{t+1} \mid x_{t+2}) \quad \text{(StepMerge Trajectory).} \tag{23}$$

We seek to bound the KL divergence between them:

$$D_{\text{KL}}(p_{\text{true}}\|p_{\text{approx}}) = \mathbb{E}_{p_{\text{true}}}\left[\log \frac{p_{\text{true}}(x_t, x_{t+1} \mid x_{t+2})}{p_{\text{approx}}(x_t, x_{t+1} \mid x_{t+2})}\right] = \mathbb{E}_{p_{\text{true}}}\left[\log \frac{\pi_\theta(x_t \mid x_{t+1})}{\pi_\theta(x_t \mid x_{t+2})}\right] \tag{24}$$

**Lemma A.2** (Timing and Value Decomposition of KL Divergence). *The KL divergence between the true and approximate distributions for consecutive timesteps decomposes into a sum of timing and value components:*

$$D_{KL}(p_{true}\|p_{approx}) = D_{KL,\ timing} + D_{KL,\ value} \tag{25}$$

*where the components are defined as:*

$$D_{KL,\ timing} = \mathbb{E}_{(S_t, x_{t+1}) \sim p_{true}}\left[\sum_{l=1}^{L} \log \frac{\tau(S_t^l \mid x_{t+1})}{\tau(S_t^l \mid x_{t+2})}\right] \tag{26}$$

$$D_{KL,\ value} = \mathbb{E}_{(S_t, V_t, x_{t+1}) \sim p_{true}}\left[\sum_{l=1}^{L} \log \frac{\nu_\theta(V_t^l \mid S_t^l, x_{t+1})}{\nu_\theta(V_t^l \mid S_t^l, x_{t+2})}\right] \tag{27}$$

*Proof.* The proof begins with the simplified expression for the KL divergence from Equation 24 and applies the timing-value factorization of Equation 19. The expectation is over the joint distribution $p_{\text{true}}(x_t, x_{t+1} \mid x_{t+2})$, where $x_t$ comprises the timing and value variables $(S_t, V_t)$.

$$D_{\text{KL}}(p_{\text{true}}\|p_{\text{approx}}) = \mathbb{E}_{(x_t, x_{t+1}) \sim p_{\text{true}}}\left[\log \frac{\pi_\theta(x_t \mid x_{t+1})}{\pi_\theta(x_t \mid x_{t+2})}\right] \tag{28}$$

$$= \mathbb{E}_{(S_t, V_t, x_{t+1}) \sim p_{\text{true}}}\left[\log \frac{\prod_l \tau(S_t^l \mid x_{t+1})\nu_\theta(V_t^l \mid S_t^l, x_{t+1})}{\prod_l \tau(S_t^l \mid x_{t+2})\nu_\theta(V_t^l \mid S_t^l, x_{t+2})}\right] \tag{29}$$

$$= \mathbb{E}_{(S_t, x_{t+1}) \sim p_{\text{true}}}\left[\sum_l \log \frac{\tau(S_t^l \mid x_{t+1})}{\tau(S_t^l \mid x_{t+2})}\right]$$

$$+ \mathbb{E}_{(S_t, V_t, x_{t+1}) \sim p_{\text{true}}}\left[\sum_l \log \frac{\nu_\theta(V_t^l \mid S_t^l, x_{t+1})}{\nu_\theta(V_t^l \mid S_t^l, x_{t+2})}\right] \tag{30}$$

Note that the timing term does not depend on the value variable $V_t$ (and only on $\tau$). Thus it can be marginalized out from the expectation over $p_{\text{true}}$. $\qquad\square$

We now seek to bound the error in timing $D_{\text{KL, timing}}$ and value separately $D_{\text{KL, value}}$.

**Definition A.3** (Conditional Mutual Information). The conditional mutual information $I(A; B \mid C)$ measures the reduction in uncertainty about random variable $A$ from knowing $B$, when $C$ is also known. It can be defined equivalently in terms of conditional entropy or as a Kullback-Leibler (KL) divergence:

$$I(A; B \mid C) = H(A \mid C) - H(A \mid B, C) = D_{\text{KL}}\big(p(a, b \mid c)\|p(a \mid c)p(b \mid c)\big). \tag{31}$$

**Assumption A.4** (Fixed Unmasking Schedule). We assume a schedule where a fixed number of $k$ tokens are unmasked at each timestep $t$.

**Lemma A.5** (Timing KL Bound). *Under the fixed unmasking schedule, the timing component of the KL divergence is bounded by the entropy of the timing decisions.*

$$D_{KL,\ timing} \le 2k \cdot \log 2 \tag{32}$$

*Proof.* The proof first establishes the equivalence between the timing KL divergence (Equation 26) and conditional mutual information, and then bounds this term using entropy.

First, we show the equivalence by applying the Markov property of the true process (Equation 22) in the numerator:

$$D_{\text{KL, timing}} = \mathbb{E}_{\tau_{\text{true}}} \left[ \log \frac{\tau(S_t \mid x_{t+1})}{\tau(S_t \mid x_{t+2})} \right] = \mathbb{E}_{\tau_{\text{true}}} \left[ \log \frac{\tau(S_t \mid x_{t+1}, x_{t+2})}{\tau(S_t \mid x_{t+2})} \right]. \tag{33}$$

We multiply and divide by $\tau(x_{t+1} \mid x_{t+2})$ to get the conditional mutual information (Theorem A.3)

$$D_{\text{KL, timing}} = \mathbb{E}_{\tau_{\text{true}}} \left[ \log \frac{\tau(S_t|x_{t+1}, x_{t+2})}{\tau(S_t|x_{t+2})} \cdot \frac{\tau(x_{t+1}|x_{t+2})}{\tau(x_{t+1}|x_{t+2})} \right] \tag{34}$$

$$= \mathbb{E}_{\tau_{\text{true}}} \left[ \log \frac{\tau(S_t, x_{t+1}|x_{t+2})}{\tau(S_t|x_{t+2})\tau(x_{t+1}|x_{t+2})} \right] \tag{35}$$

$$= I(S_t; x_{t+1} \mid x_{t+2}). \tag{36}$$

We bound the conditional mutual information term by relating it to the entropy of the timing decisions, which can be decomposed on a per-token basis.

$$\begin{aligned}
I(S_t; x_{t+1} \mid x_{t+2}) &= H(S_t \mid x_{t+2}) - H(S_t \mid x_{t+1}, x_{t+2}) &&\text{(by definition of mutual information)} \\
&\leq H(S_t \mid x_{t+2}) &&\text{(since entropy is non-negative)} \\
&= \sum_{l=1}^{L} H(S_t^l \mid x_{t+2}) &&\text{(by cond. independence of tokens)}
\end{aligned} \tag{37}$$

To evaluate this sum, we partition the tokens into those that unmask in the $[t, t+2)$ interval and those that do not.

$$\begin{aligned}
\sum_{l=1}^{L} H(S_t^l \mid x_{t+2}) &= \sum_{l \in \text{Unmasked}} H(S_t^l \mid x_{t+2}) + \sum_{l \notin \text{Unmasked}} H(S_t^l \mid x_{t+2}) &&\text{(by partitioning the sum)} \\
&\leq \sum_{l \in \text{Unmasked}} \log 2 + 0 &&\text{(by bounding each term)} \\
&= 2k \cdot \log 2 &&\text{(by summing over the set)}
\end{aligned} \tag{38}$$

The second sum vanishes as its tokens have a deterministic timing decision ($S_t^l = 0$), resulting in zero entropy. The first sum is over the $2k$ tokens that unmask in the interval. For each token, $S_t^l$ is a binary random variable representing the choice of unmasking at time $t$ or $t+1$. The entropy of such a variable is maximized at $\log 2$ under maximum uncertainty (a uniform distribution over the two outcomes). Using this upper bound for each of the $2k$ tokens yields the final result. $\qquad \square$

**Definition A.6** (Value Prediction Sensitivity). Let $\epsilon$ be the maximum per-token, pointwise log-ratio of value probabilities between consecutive timesteps, maximized over all possible values, tokens, and states where an unmasking occurs.

$$\epsilon = \max_{v,l,x_{t+1},x_{t+2}} \log \frac{\nu_\theta(V_t^l = v \mid S_t^l = 1, x_{t+1})}{\nu_\theta(V_t^l = v \mid S_t^l = 1, x_{t+2})} \tag{39}$$

$$= \max_{v,l,x_{t+1},x_{t+2}} \log \frac{\text{softmax}(f_\theta(x_{t+1})_l)_v}{\text{softmax}(f_\theta(x_{t+2})_l)_v} \tag{40}$$

$$= \max_{v,l,x_{t+1},x_{t+2}} \left( \underbrace{f_\theta(x_{t+1})_{l,v} - f_\theta(x_{t+2})_{l,v}}_{\text{Logit Difference}} - Z_l(x_{t+1}, x_{t+2}) \right) \tag{41}$$

for the difference between log-softmax normalizers $Z_l(x_{t+1}, x_{t+2}) = \log \left( \frac{\sum_j \exp\left(f_\theta(x_{t+1})_{l,j}\right)}{\sum_j \exp\left(f_\theta(x_{t+2})_{l,j}\right)} \right)$.

**Remark A.7** (Interpetation of Value Prediction Sensitivity). *The value prediction sensitivity $\epsilon$ provides a worst-case guarantee on the stability of the model's value distribution $\nu_\theta$ between consecutive timesteps. It bounds how much $\nu_\theta$ can change for any token value $v$ at a position $l$ by constraining fluctuations in the underlying neural network's raw logits—namely the difference between $f_\theta(\mathbf{x}_{t+1})_{l,v}$ and $f_\theta(\mathbf{x}_{t+2})_{l,v}$ after normalization. A small $\epsilon$ therefore signifies that the entire logit vector for position $l$ remains relatively constant when the context shifts by one step (from $x_{t+1}$ to $x_{t+2}$). This logit-level stability ensures that the model's predictive distribution is not volatile, thereby validating our StepMerge approximation, which relies on the assumption that intermediate states can be skipped without drastically altering the final trajectory likelihood.*

**Lemma A.8** (Value KL Bound). *Under the fixed unmasking schedule, the value component of the KL divergence is bounded in terms of the value prediction sensitivity $\epsilon$.*

$$D_{KL,\,value} \leq k \cdot \epsilon \tag{42}$$

*Proof.* We start with the definition of the value KL divergence:

$$D_{\text{KL, value}} = \mathbb{E}_{p_{\text{true}}} \left[ \sum_{l=1}^{L} \log \frac{\nu_\theta(V_t^l \mid S_t^l, x_{t+1})}{\nu_\theta(V_t^l \mid S_t^l, x_{t+2})} \right] \tag{43}$$

To analyze the sum inside the expectation, we partition the token indices based on the timing decision $S_t^l$, which indicates whether token $l$ is unmasked at step $t$. Let $\mathcal{U}_t = \{l \mid S_t^l = 1\}$ be the set of indices for tokens that unmask, and $\mathcal{U}_t^c = \{l \mid l \notin \mathcal{U}_t\} = \{l \mid S_t^l = 0\}$ be the complementary set for all other tokens. The sum can be explicitly decomposed as:

$$\sum_{l=1}^{L} \log \frac{\nu_\theta(\dots)}{\nu_\theta(\dots)} = \underbrace{\sum_{l \in \mathcal{U}_t} \log \frac{\nu_\theta(V_t^l \mid S_t^l = 1, x_{t+1})}{\nu_\theta(V_t^l \mid S_t^l = 1, x_{t+2})}}_{\text{Tokens unmasked at time } t} + \underbrace{\sum_{l \in \mathcal{U}_t^c} \log \frac{\nu_\theta(V_t^l \mid S_t^l = 0, x_{t+1})}{\nu_\theta(V_t^l \mid S_t^l = 0, x_{t+2})}}_{\text{Tokens } not \text{ unmasked at time } t} \tag{44}$$

$$\leq \sum_{l \in \mathcal{U}_t} \epsilon + \sum_{l \in \mathcal{U}_t^c} \log \frac{1}{1} \tag{45}$$

$$= k \cdot \epsilon \tag{46}$$

For tokens that are unmasked at time $t$ ($l \in \mathcal{U}_t$), the value distribution $\nu_\theta$ is defined by the model's softmax output, and the log-ratio is bounded by the value prediction sensitivity ($\epsilon$). For any token that is not unmasked at time $t$ ($l \in \mathcal{U}_t^c$), the value-setting process is a deterministic identity transformation, where $V_t^l = x_{t+1}^l$ with probability 1.

Since this upper bound holds for any trajectory, the expectation over all trajectories is also bounded by the same constant. $\square$

**Lemma A.9** (Consecutive Timestep Bound). *Under the fixed unmasking schedule Theorem A.4, the total KL divergence between the true and approximate distributions for consecutive timesteps is bounded by the sum of the timing and value bounds.*

$$D_{KL}(p_{true}\|p_{approx}) \leq 2k \cdot \log 2 + k \cdot \epsilon \tag{47}$$

*Proof.* The result follows directly by combining the bounds from the preceding lemmas.

$$D_{\text{KL}}(p_{\text{true}}\|p_{\text{approx}}) = D_{\text{KL, timing}} + D_{\text{KL, value}} \leq (2k \cdot \log 2) + (k \cdot \epsilon) \tag{48}$$

$\square$

### A.2.3 ENTIRE TRAJECTORY BOUND

We extend the analysis from an $L$-length sequence at consecutive timesteps $x_t, x_{t+1}$ to the entire trajectory $x_0 \dots x_T$ with $N$ total blocks. To do so, first consider block $n$ with $B = L/N$ timesteps spanning $t = [nB, nB+1, \dots, nB+(B-1)]$.

Within this block, the true reverse process is a Markov chain over the full trajectory:

$$p_{\text{true}}^{(n)}(x_{nB}, \ldots, x_{nB+(B-1)} \mid x_{nB+B}) = \prod_{t=nB}^{nB+(B-1)} \pi_\theta(x_t \mid x_{t+1}) \quad \text{(Full Trajectory)} \tag{49}$$

The StepMerge approximation, however, assumes each state $x_t$ in the block is generated independently conditioned only on the final state $x_{nB+B}$:

$$p_{\text{approx}}^{(n)}(x_{nB}, \ldots, x_{nB+(B-1)} \mid x_{nB+B}) = \prod_{t=nB}^{nB+(B-1)} \pi_\theta(x_t \mid x_{nB+B}) \quad \text{(StepMerge Trajectory)} \tag{50}$$

**Lemma A.10** (Block KL Bound). *Under the fixed unmasking schedule Theorem A.4, the KL divergence for block $n$ is bounded by:*

$$D_{KL}^{(n)}(p_{true}^{(n)} \| p_{approx}^{(n)}) \leq \frac{kB(B+1)}{2} \log 2 + Bk \cdot \epsilon_{block} \tag{51}$$

*where $\epsilon_{block}$ is the maximum value prediction sensitivity over the timesteps in the block.*

*Proof.* The KL divergence for block $n$ is the expectation of the log-ratio of the true and approximate distributions.

$$D_{\text{KL}}^{(n)} = \mathbb{E}_{p_{\text{true}}^{(n)}} \left[ \log \frac{\prod_{t=nB}^{nB+(B-1)} \pi_\theta(x_t \mid x_{t+1})}{\prod_{t=nB}^{nB+(B-1)} \pi_\theta(x_t \mid x_{nB+B})} \right] \tag{52}$$

$$= \mathbb{E}_{p_{\text{true}}^{(n)}} \left[ \sum_{t=nB}^{nB+(B-1)} \log \frac{\pi_\theta(x_t \mid x_{t+1})}{\pi_\theta(x_t \mid x_{nB+B})} \right] \tag{53}$$

$$= \underbrace{\mathbb{E}_{p_{\text{true}}^{(n)}} \left[ \sum_{t=nB}^{nB+(B-1)} \log \frac{\tau(S_t \mid x_{t+1})}{\tau(S_t \mid x_{nB+B})} \right]}_{D_{\text{KL, timing}}^{(n)}} + \underbrace{\mathbb{E}_{p_{\text{true}}^{(n)}} \left[ \sum_{t=nB}^{nB+(B-1)} \log \frac{\nu_\theta(v_t \mid S_t, x_{t+1})}{\nu_\theta(v_t \mid S_t, x_{nB+B})} \right]}_{D_{\text{KL, value}}^{(n)}} \tag{54}$$

Using the timing-value factorization Equation 19 and linearity of expectation, we decompose into block-level timing and value components, $D_{\text{KL}}^{(n)} = D_{\text{KL, timing}}^{(n)} + D_{\text{KL, value}}^{(n)}$, which we bound separately.

**Timing Bound.** The timing component is the sum of conditional mutual information terms

$$D_{\text{KL, timing}}^{(n)} = \sum_{t=nB}^{nB+(B-1)} \mathbb{E}_{p_{\text{true}}^{(n)}} \left[ \log \frac{\tau(S_t \mid x_{t+1})}{\tau(S_t \mid x_{nB+B})} \right] = \sum_{t=nB}^{nB+(B-1)} I(S_t; x_{t+1} \mid x_{nB+B}) \tag{55}$$

following the derivation in Theorem A.5. We can further bound the conditional mutual information by the conditional entropy from Theorem A.3:

$$I(S_t; x_{t+1} \mid x_{nB+B}) = H(S_t \mid x_{nB+B}) - H(S_t \mid x_{t+1}, x_{nB+B}) \leq H(S_t \mid x_{nB+B}). \tag{56}$$

Due to mutual exclusivity, each token can unmask at most once. For a token $l$ masked at $x_{nB+B}$, define $T_l$ as its unmasking time (if any). The collection $\{S_t^l\}_{t=nB}^{nB+B-1}$ encodes which value $T_l$ takes, where $T_l \in \{nB, ..., nB+B-1, \infty\}$. Thus:

$$\sum_{t=nB}^{nB+B-1} H(S_t^l \mid x_{nB+B}) \leq H(T_l \mid x_{nB+B}^l = \mathbf{m}) \leq \log(B+1) \tag{57}$$

Since exactly $kB$ tokens unmask during the block:

$$D_{\text{KL, timing}}^{(n)} \leq \sum_{l \in \mathcal{M}_{nB+B}} H(T_l \mid x_{nB+B}^l = \mathbf{m}) \leq kB \cdot \log(B+1) \tag{58}$$

where $\mathcal{M}_{nB+B} = \{l : x_{nB+B}^l = \mathbf{m}\}$ is the set of masked tokens at the block end.

**Value Bound.** The value component of the divergence is:

$$D_{\text{KL, value}}^{(n)} = \sum_{t=nB}^{nB+(B-1)} \mathbb{E}_{p_{\text{true}}^{(n)}} \left[ \sum_{l=1}^{L} \log \frac{\nu_\theta(V_t^l \mid S_t^l, x_{t+1})}{\nu_\theta(V_t^l \mid S_t^l, x_{nB+B})} \right] \tag{59}$$

Following Theorem A.6, we define a block-level value prediction sensitivity $\epsilon_{block}$ as the maximum log-ratio within the block:

$$\epsilon_{block} = \max_{t\in[nB,(n+1)B),v,l,x_{t+1}} \log \frac{\nu_\theta(V_t^l = v \mid S_t^l = 1, x_{t+1})}{\nu_\theta(V_t^l = v \mid S_t^l = 1, x_{nB+B})} \tag{60}$$

At each timestep $t$, the inner sum over tokens $l$ is non-zero only for the $k$ tokens being unmasked ($S_t^l = 1$) by definition of $\nu_\theta$ (Equation 21). For each of these, the log-ratio is bounded by definition by $\epsilon_{block}$. Therefore, the entire sum inside the expectation is bounded by $k \cdot \epsilon_{block}$. Summing over the $B$ timesteps yields the total value bound:

$$D_{\text{KL, value}}^{(n)} \leq \sum_{t=nB}^{nB+(B-1)} k \cdot \epsilon_{block} = Bk \cdot \epsilon_{block} \tag{61}$$

**Total Bound.** Combining the bounds for the timing and value components gives the final result for the block-level KL divergence.

$$D_{\text{KL}}^{(n)} = D_{\text{KL, timing}}^{(n)} + D_{\text{KL, value}}^{(n)} \leq kB \cdot \log(B+1) + Bk \cdot \epsilon_{block} \tag{62}$$

$\square$

Finally, we aggregate the per-block errors to establish a bound for the entire generation trajectory. Since the approximation for each block is conditionally independent of the others given the state at the end of the block, the total KL divergence is the sum of the per-block KL divergences.

**Theorem A.11** (Main Bound). *Let $L$ be the total number of tokens (each unmasked exactly once), $B = T/N$ be the number of timesteps per block, and $\epsilon_{block}$ be the maximum value prediction sensitivity within a block. The KL divergence between the true sequential process (Full Trajectory) and the block-parallel approximation (StepMerge Trajectory) is bounded by:*

$$D_{KL}(p_{true}\|p_{approx}) \leq L \cdot \log(B+1) + L \cdot (B-1) \cdot \epsilon_{block} \tag{63}$$

*Proof.* For block $n$, let $k$ be the number of tokens unmasked per timestep in that block. From the tighter timing bound using mutual exclusivity and the value bound:

$$D_{\text{KL}}^{(n)} \leq kB \cdot \log(B+1) + kB \cdot \epsilon_{block} \tag{64}$$

Summing over all $N$ blocks:

$$D_{\text{KL}} = \sum_{n=0}^{N-1} D_{\text{KL}}^{(n)} \leq \sum_{n=0}^{N-1} \left( kB \cdot \log(B+1) + kB \cdot \epsilon_{block} \right) \tag{65}$$

$$= \left( \sum_{n=0}^{N-1} kB \right) \log(B+1) + \left( \sum_{n=0}^{N-1} kB \right) \epsilon_{block} \tag{66}$$

Since each token unmasks exactly once and there are $L$ tokens total we know $\sum_{n=0}^{N-1} kB = L$. This yields:

$$D_{\text{KL}} \leq L \cdot \log(B+1) + L \cdot \epsilon_{\text{block}} \tag{67}$$

Substituting $B = T/N$:

$$D_{\text{KL}} \leq L \cdot \log \left( \frac{T}{N} + 1 \right) + L \cdot \epsilon_{\text{block}} \tag{68}$$

$\square$

### A.3 PROOF OF THEOREM 4.3

*Proof.* We first prove the *necessity*. If a DLM policy $\pi_\theta$ is any-order causal, then by the one-shot evaluation trick described in Section 3.2.2, the likelihood of an entire trajectory can be computed with a single forward pass of the model.

We now prove the *sufficiency* by contradiction. Suppose a *reasonable* $\pi_\theta$ is *not* any-order causal. Then at least one of the three defining conditions fails. We check them one by one.

**Condition 1.** Assume there exist $1 \leq l \leq L$ and distinct $0 \leq t_i, t_j < t_l$ such that $A_{t_i}^l \neq A_{t_j}^l$. By the sampling process, there are mask tokens $\mathbf{m}_{t_i}^{l_{t_i}}$ and $\mathbf{m}_{t_j}^{l_{t_j}}$ newly unmasked at $t_i$ and $t_j$, respectively. The reasonable DLM assumption gives $l \in A_{t_i}^{l_{t_i}}$ and $l \in A_{t_j}^{l_{t_j}}$. Hence $\mathbf{m}_{t_i}^{l_{t_i}}$ and $\mathbf{m}_{t_j}^{l_{t_j}}$ both attend to $\mathbf{x}^l$ when decoded, yet $\mathbf{x}^l$ attends to different token sets at $t_i$ and $t_j$. Thus their likelihoods cannot be jointly evaluated in one forward pass — a contradiction. So Condition 1 must hold.

**Condition 2.** Assume there exists $1 \leq l \leq L$ with $A_{t_l}^l \nsubseteq \Omega_l \cup \{l\}$.

Since $\Omega_l \subseteq A_{t_l}^l$, there is some other position $l'$ so that at time $t_l$ the token at $l'$ is still a mask $\mathbf{m}^{l'}$. Then $\mathbf{m}^l$ attends to $\mathbf{m}^{l'}$ when decoded. By the reasonable DLM assumption we also have $l \in A_{t_{l'}}^{l'}$. So the query token $\mathbf{m}^{l'}$ knows what the clean token decoded at position $l$ is. Then, when we do the one model forward pass, the information of clean token $l$ flows into the query embedding of $\mathbf{m}^{l'}$, and since there are more than one layers in the Transformer network, this information further flows into the key value embeddings of $\mathbf{m}^{l'}$, which $\mathbf{m}^l$ attends to. This creates information leakage because $\mathbf{m}^l$ will be exposed to the information of its clean token in the one-model forward pass evaluation. Thus, Condition 2 must hold.

**Condition 3.** Assume there exists $1 \leq l \leq L$ such that $\mathbf{x}^l$ is not among the last two decoded tokens, and some $t < t_l$ with $A_t^l \nsubseteq \Omega_l \cup U_l$. Then there is a position $l'$ with $l' \notin \Omega_l \cup U_l$ but $l' \in A_t^l$. According to Condition 1, $l' \in A_t^l$ is true for all $0 \leq t < t_l$. Two possibilities arise:

1. $l'$ **is the position decoded right after** $l$**.** Since $\mathbf{x}^l$ is not among the last two decoding steps, there exist at least one token decoded after token $l'$ (denoted as $l''$). According to the reasonable DLM assumption, token $l'$ and token $l''$ both have to attend to $\mathbf{x}^l$. However, the token value at position $l'$ must be different at time $t_{l'}$ and $t_{l''}$, making it impossible to build the attention mask for query token $\mathbf{x}^l$ in the one forward pass setting.

2. $l'$ **is not the position decoded right after** $l$**.** There must be one mask token decoded in between $l$ and $l'$ (denoted as $l''$). By the reasonable DLM assumption, $\mathbf{m}^{l''}$ should attend to $\mathbf{x}^l$ when $\mathbf{m}^{l''}$ is decoded, and $\mathbf{x}^l$ should attend to $\mathbf{m}^{l'}$. Thus, $\mathbf{m}^{l''}$ indirectly depends on $\mathbf{m}^{l'}$ at time step $t_{l''}$, violating Condition 2.

In either subcase, one-shot evaluation fails. So Condition 3 must also hold.

In conclusion, since violating any of the conditions leads to a contradiction, the three conditions are jointly necessary and sufficient. □

## B D2 IMPLEMENTATION DETAILS

### B.1 ANY-ORDER DECODING THAT ALLOWS PARALLEL GENERATION.

We present Figure 10 to illustrate how our any-order decoding pattern allows parallel generation.

### B.2 ANY-ORDER DECODING WITH PROMPT

In this section, we explicitly introduce the any-order decoding algorithm for sequences with a prompt to promote understanding of our method. Here, we slightly change the notation to denote the prompt as $\mathbf{q}^{1:\mathbf{L_P}}$, where each $\mathbf{q}^l$ represents the prompt token at the $l_{th}$ position. Starting from $\mathbf{q}^{1:\mathbf{L_P}} \oplus \mathbf{m}^{\mathbf{L_P}+1:\mathbf{L_P}+\mathbf{L}}$, we specify the attention pattern of each token at all the time steps. First, we

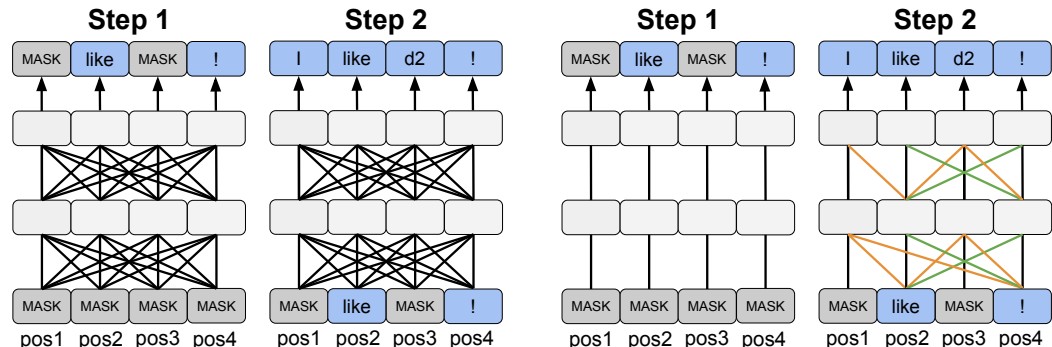

(a) Fully bidirectional MDLM decoding.    (b) Any-order decoding.

Figure 10: Illustration of different DLM decoding strategies when more than one tokens are decoded at a single time step. We depict attention with query tokens (one layer up) attending to keys/values (one layer below) via an undirected connected line. The output at each position is depicted with a directed arrow. "pos" refers to positional encoding index. We use a four token example where the decoding order is "like,!$\rightarrow$ I, d2". At each time step, newly added attention relations in any-order decoding are highlighted. There are two types of new attention patterns: new masked to unmasked attentions are highlighted using red markers, and new unmasked to unmasked patterns are highlighted using green markers.

fix the prompt tokens' attention pattern as $\forall 1 \leq l \leq L_P, attn_t^l = \{l^p \mid 1 \leq l^p \leq L_P\}$. Next, for the tokens that actually get decoded, a mask token $\mathbf{m}^l$ attends to $\mathbf{q}^{1:L_P} \cup \mathbf{x}_0^{\Omega_l} \cup \{\mathbf{m}^l\}$ when it is decoded. After it is decoded as $\mathbf{x}_0^l$, it attends to $\mathbf{q}^{1:L_P} \cup \mathbf{x}_0^{\Omega_l} \cup \mathbf{x}^{U_{t_l}}$, and this attention pattern stays unchanged for the rest of the sampling process.

### B.3 ONE-SHOT TRAJECTORY LIKELIHOOD

Following Appendix B.2, we introduce the one-shot trajectory likelihood evaluation for sequences with a prompt. In this case, a sampled trajectory of $L$ tokens is denoted as $\mathbf{x}_{0:T}^{L_P+1:L_P+L}$. Similarly to Section 4.2, if it is generated using any-order decoding, its trajectory likelihood $\pi_\theta(\mathbf{x}_{0:T}^{L_P+1:L_P+L} \mid \mathbf{q}^{1:L_P}) = \prod_{t=0}^{T-1} \prod_{l \in U_t} \pi_\theta(\mathbf{x}_t^l \mid \mathbf{x}_{t+1}^{L_P+1:L_P+L}, \mathbf{q}^{1:L_P})$ can be attained with one model pass.

Concretely, we first build a $L_P + 2L$ sequence $\mathbf{q}^{1:L_P} \oplus \mathbf{x}_0^{1:L} \oplus \mathbf{m}^{L+1:2L}$, where $\oplus$ indicates concatenation along the sequence dimension. The positional encoding indices $\text{pos}_l$ are assigned as follows:

$$\text{pos}_l = \begin{cases} l, & l \leq L_P + L \\ l - L, & L_P + L < l \leq L_P + 2L \end{cases} \tag{69}$$

We then craft the attention mask so that as a query token, $\mathbf{q}^l$ attends to $\mathbf{q}^{1:L_P}$, $\mathbf{x}_0^l$ attends to $\mathbf{q}^{1:L_P} \cup \mathbf{x}_0^{\Omega_l} \cup \{\mathbf{x}_0^{U_{t_l}}\}$, and $\mathbf{m}^{L+l}$ attends to $\mathbf{q}^{1:L_P} \cup \mathbf{x}_0^{\Omega_l} \cup \{\mathbf{m}^{L+l}\}$. In Figure 11, we provide an example of the attention mask pattern for both any-order decoding and the one-shot likelihood evaluation.

## C D2 TRAINING ALGORITHMS

We present the pseudocode of d2-StepMerge and d2-AnyOrder in Algorithm 1 and Algorithm 2.

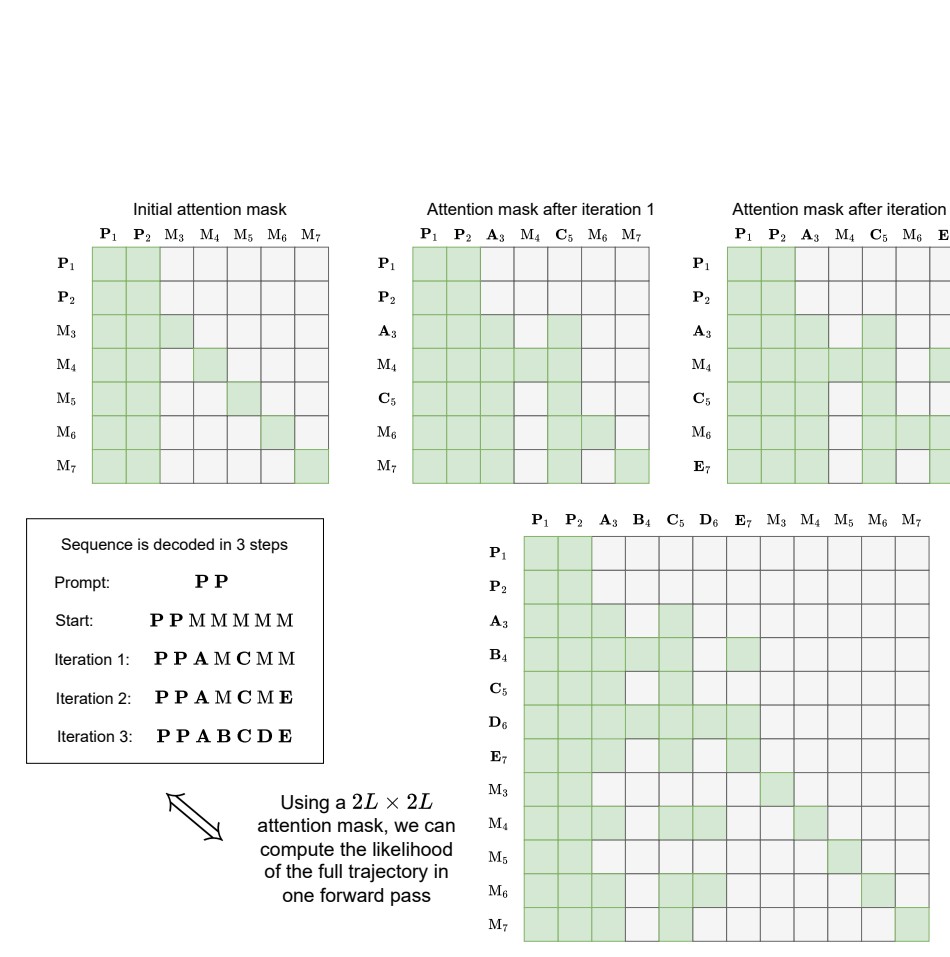

Figure 11: Attention masks that respect the causal order of generation. Tokens can only attend to prompt tokens, to themselves, and to tokens that were decoded at an early iteration. Green squares represent attention is 'turned on', i.e., attention bias of $0$, between queries (rows) and keys (columns), gray represents no attention, i.e., attention bias of $-\infty$. Subscripts denote positional embedding ids. In this example, the sequence is generated as a completion to the prompt **PP** with the following trajectory: $\text{MMMMM} \rightarrow \textbf{A}\text{M}\textbf{C}\text{MM} \rightarrow \textbf{A}\text{M}\textbf{C}\text{M}\textbf{E} \rightarrow \textbf{ABCDE}$. *(Top)* Any-order causal attention mask during generation. *(Bottom)* $2L \times 2L$ attention mask that enables computation of likelihood for entire sequence trajectory in one forward pass. The model receives a $2L$ sequence consisting of a concatenation of the final $L$ generated tokens and $L$ mask tokens. The likelihood is computed using the output of the second $L$ output tokens.

## D  LLM USAGE

**Use of Large Language Models.** LLMs were used only as an editorial tool to refine the writing style and enhance clarity. No text, formulas, algorithms, or experimental results were generated or suggested by LLMs; all scientific contributions are original work by the authors.

---

**Algorithm 1** d2-StepMerge

---

**Input:** Reward model $r$, reference model $\pi_{\text{ref}}$, prompt distribution $\mathcal{Q}$, number of completions per prompt $G$, number of inner updates $n$, number of time segments $N$.

Initialize $\pi_\theta \leftarrow \pi_{\text{ref}}$

**repeat**

    $\pi_{\text{old}} \leftarrow \pi_\theta$

    Sample prompt $\mathbf{q} \sim \mathcal{Q}$

    Sample $G$ completion trajecotries $\{\mathbf{x}_{0:T}^{(i)}\}_{i=1}^G \sim \pi_{\text{old}}(\cdot \mid \mathbf{q})$

    Compute advantage $\{A_{(i)}\}_{i=1}^G$ (see Section 2.2)

    **for** $j$ 1 to $N$ **do**

        `stop_gradient`$(\text{Compute and collect } \{\pi_{\text{old}}(\mathbf{x}_{j\frac{T}{N}:(j+1)\frac{T}{N}}^{(i)} \mid \mathbf{q})\}_{i=1}^G)$

        `stop_gradient`$(\text{Compute and collect } \{\pi_{\text{ref}}(\mathbf{x}_{j\frac{T}{N}:(j+1)\frac{T}{N}}^{(i)} \mid \mathbf{q})\}_{i=1}^G)$

    **end for**

    **for** gradient_iterations 1 to $n$ **do**

        **for** $j$ 1 to $N$ **do**

            Compute d2-StepMerge GRPO objective (Eq. (6)) with respect to $\{\mathbf{x}_{j\frac{T}{N}:(j+1)\frac{T}{N}}^{(i)}\}_{i=1}^G$

            Backward pass to calculate gradient

        **end for**

        Update $\theta$ with optimizer.

    **end for**

**until** converged

**return** $\pi_\theta$

---

---

**Algorithm 2** d2-AnyOrder

---

**Input:** Reward model $r$, reference model $\pi_{\text{ref}}$, prompt distribution $\mathcal{Q}$, number of completions per prompt $G$, number of inner updates $n$, number of sampling time steps $T$.

Initialize $\pi_\theta \leftarrow \pi_{\text{ref}}$

**repeat**

    $\pi_{\text{old}} \leftarrow \pi_\theta$

    Sample prompt $\mathbf{q}^{1:L_P} \sim \mathcal{Q}$

    Sample $G$ completions $\{\mathbf{x}_0^{L_P+l:L_P+L_{(i)}}\}_{i=1}^G \sim \pi_{\text{old}}(\cdot \mid \mathbf{q}^{1:L_P})$

    Compute advantage $\{A_{(i)}\}_{i=1}^G$ (see Section 2.2)

    Build input sequence $INPUT = \mathbf{q}^{1:L_P} \oplus \mathbf{x}_0^{L_P+1:L_P+L_{(i)}} \oplus \mathbf{m}^{L_P+1:L_P+2L_{(i)}})\}_{i=1}^G$

    Build attention mask, see Appendix B.3

    `stop_gradient`$(\text{compute } \{\pi_{\text{old}}^{\text{AO}}(\mathbf{x}_0^{L_P+1:L_P+L_{(i)}} \mid INPUT\})$

    `stop_gradient`$(\text{compute } \{\pi_{\text{ref}}^{\text{AO}}(\mathbf{x}_0^{L_P+1:L_P+L_{(i)}} \mid INPUT\})$

    **for** gradient_iterations 1 to $n$ **do**

        Compute d2-AnyOrder GRPO objective (Eq. (6))

        Update $\theta$ with optimizer.

    **end for**

**until** converged

**return** $\pi_\theta$

---

