# OpenReview forum: "d2: Improved Techniques for Training Reasoning Diffusion Language Models"
_ICLR.cc/2026/Conference — Submitted to ICLR 2026_

### Official Review · Reviewer_QLxB · 2025-10-25

**Soundness:** 2
**Presentation:** 2
**Contribution:** 3
**Rating:** 4
**Confidence:** 5

**Summary:**

This paper introduces d2, a new policy optimization framework designed to enhance the reasoning capabilities of masked diffusion language models. The core contribution is a novel GRPO-style policy gradient algorithm that relies on two new estimators for the computationally intractable trajectory likelihood. The first, d2-StepMerge, is a practical, biased estimator that evaluates the likelihood at $N$ discrete steps, providing a theoretically-analyzed trade-off between compute and accuracy that works on existing models like LLaDA. The second, d2-AnyOrder, is a highly efficient, unbiased, single-pass estimator, but it requires a specific "any-order causal" property which the paper formally defines and shows is absent in standard DLMs. Empirically, d2 (using the StepMerge estimator) sets a new state-of-the-art for DLMs on several reasoning benchmarks.

**Strengths:**

The most significant finding is that the proposed d2 framework achieves state-of-the-art results on multiple reasoning benchmarks (e.g., a massive jump to 91.9% on Sudoku) using reinforcement learning without any supervised fine-tuning, demonstrating a more scalable path for imbuing DLMs with reasoning. The authors originally derive a GRPO-style policy gradient for DLMs and introduce two novel likelihood estimators, d2-StepMerge and d2-AnyOrder, to solve the intractable likelihood problem. The paper features a rigorous theoretical derivation (Theorem 3.1), a formal bound on the d2-StepMerge approximation error (Theorem 4.1), and thorough, FLOPs-based empirical validation. Furthermore, the paper has intuitive diagrams (Figure 2) that makes the content more accessible.

**Weaknesses:**

1. The inclusion of the d2-anyorder estimator and its related "any-order causality" theory detracts from the paper's core contribution. The primary, SOTA-achieving results are based entirely on d2-stepmerge applied to LLaDA. The d2-anyorder section uses a different model, task, and baseline, which fragments the narrative, confuses the central message, and reduces the paper's overall focus and clarity.

2. The analysis of the d2-stepmerge compute-bias tradeoff is incomplete. While Figure 7 explores various values of $N$, it critically omits the $N=T$ (full trajectory, no-merge) baseline. This data point is essential to represent the "zero-bias" anchor. Without it, it is difficult to fully quantify the "bias" being traded off or to know if the chosen $N=16$ has fully converged to the best possible performance achievable by the underlying policy gradient.

3. The RL objective is correctly defined over the final sequence likelihood $x_0$ (Eq. 3) (which is intractable for sure), as this is what the reward function $r(x_0)$ evaluates. However, the final policy update (Eq. 5) is expressed in terms of the likelihood of the full diffusion trajectory $x_{0:T}$. The main text fails to adequately explain why optimizing the trajectory likelihood is a valid or equivalent substitute for optimizing the final sequence likelihood, making the theoretical leap from the stated objective to the final algorithm feel abrupt and unmotivated.

4. The motivation for using an off-policy algorithm like GRPO is fundamentally flawed. The key benefit of GRPO/PPO is rollout data reuse, which relies on cheap likelihood calculations when calculating importance ratio. In this DLM setting, calculating the importance ratio requires $N$ expensive forward passes, negating this advantage. This makes the off-policy updates computationally expensive, and it may even be less efficient than a simple fully on-policy algorithm. The paper fails to justify this choice or provide a comparison against an on-policy baseline.

**Questions:**

1. There appears to be a typo in Eq. 6, which defines the d2-StepMerge objective. The time indices for the diffusion states are written as $nL/N$ and $(n+1)L/N$, using the sequence length $L$. However, the text describes partitioning the $T$ diffusion time steps into $N$ segments. Should these indices not be $nT/N$ and $(n+1)T/N$ respectively?

2. I would like to know more detailed explanation of how $D_N$ is computed in Figure 3. Eq. 7 defines $D_N$ as the KL divergence between the full T-step trajectory distribution and the N-step (StepMerge) approximate distribution. Since computing the full KL divergence is computationally expensive—requiring expectations over all possible tokens—it would be helpful to clarify how the KL divergence is actually estimated in practice.

**Details Of Ethics Concerns:**

The paper does not appear to raise any ethical concerns. It does not involve human subjects, sensitive data, or potentially harmful applications.

---

### Official Review · Reviewer_M9go · 2025-10-31

**Soundness:** 2
**Presentation:** 2
**Contribution:** 2
**Rating:** 4
**Confidence:** 4

**Summary:**

d2 provides an RL method to reinforce the whole denoising trajectory in MDLMs. The authors identify the computational challenge of estimating trajectory likelihoods in non-autoregressive models as a key bottleneck for applying RL. To address this, they propose two distinct policy gradient estimators: d2-StepMerge, which trades off computation for accuracy by merging timesteps, and d2-AnyOrder, which enables an unbiased, one-shot likelihood estimate for models satisfying a specific "any-order causality" property. The authors conduct experiments to validate the effectiveness of d2 on LLaDA and Eso-LM.

**Strengths:**

1. I agree that the whole generation trajectory in MDLMs should be fully taken into consideration in RL algorithms, and StepMerge provides a practical way for ease training overload.
2. d2-AnyOrder is an ideal way for any-order generation, which can make use of KV-Cache, and need only one pass for probability estimation.
3. The paper provides extensive experiments across multiple reasoning tasks Sudoku, Countdown, GSM8K, MATH. The ablation studies on the number of segments $N$ are particularly insightful to demonstrate the proper value range for $N$

**Weaknesses:**

1. StepMerge merges adjacent steps into a segment so that the number of model passes are reduced. However, why don't we just sample the trajectory with fewer steps, and just reinforce such shorter trajectories? I think that is necessary for comparison.
2. d2-anyorder is ideal. However, it only applies to models that already have such generation ability. It cannot be applied to popular models such as LLaDA, Dream.
3. There are too few comparisons, such as other MDLM-related RL-methods [1] or more up-to-date MDLMs [2].

[1] Reinforcing the Diffusion Chain of Lateral Thought with Diffusion Language Models

[2] Dream 7B: Diffusion Large Language Models

**Questions:**

Here are some experiment details I would like to know:
1. I realize that the computation graph of a whole trajectory could be too heavy to be preserved in GPU. Could you provide more details on LLaDA experiments? Such as: the generation length, the number of steps for results in Tab.2.
2. d1 and wd1 are actually trained with LoRA. However, the experiment part does not mention the training settings. Are results in Tab. 2 trained with LoRA or full parameters? Are they compared fairly in the same setting?

---

### Official Review · Reviewer_fryY · 2025-11-02

**Soundness:** 3
**Presentation:** 3
**Contribution:** 2
**Rating:** 4
**Confidence:** 4

**Summary:**

This paper proposes a clean theoretical framework for applying RL to diffusion language models. The key contribution is d2-StepMerge, which approximates trajectory log probabilities by evaluating likelihood at N merged denoising steps instead of all T timesteps, offering a  trade-off between computational cost and approximation accuracy. The paper showing that approximation error decreases as N increases. Additionally, the paper introduces d2-AnyOrder, which enables one-shot trajectory likelihood evaluation for any-order causal diffusion models, which is novel. The method achieves significant performance improvements over existing baselines (d1, wd1) across reasoning benchmarks.

**Strengths:**

1. Applying RL to any-order causal diffusion LLMs is novel and represents a solid contribution, as prior work has only applied RL to fully-bidirectional diffusion models like LLaDA (d1, wd1) or block-based diffusion approaches like TraceRL. And it shows potential of any-order dLLMs as it can get log prob with one forward pass without approximation with designed attention mask.
2. The d2-StepMerge method is well-motivated and demonstrates strong empirical performance, outperforming previous sota methods across all four reasoning benchmarks.
3. The paper provides a clean, theoretically grounded formalization of RL for diffusion LLMs, deriving the objective from first principles.
4. The thorough analysis of the computation-performance trade-off is valuable, as d2 methods necessarily require more computation per update than baselines like d1 (N=1). The paper transparently shows that this additional cost is justified by superior performance and provides guidance for selecting N in practice.

**Weaknesses:**

My main concern is that the paper mainly concerns P(response | prompt, denoising_order) for a specific, human-selected denoising order, treating this conditional likelihood as the training objective. However, the true quantity we care about is P(response | prompt), which should marginalize over all possible denoising orders. Since this marginalization is intractable, the paper effectively assumes that optimizing for one fixed order generalizes to the overall response quality.

This might introduces a potential approximation error: when a response is bad, we ideally want the model to avoid generating it under any denoising order, but the current approach only discourages it under the specific order used during training. Different denoising orders could lead to the same final response through different intermediate paths, and the model might learn order-specific artifacts rather than genuinely improving response quality. The paper can be improved if the authors can discuss if this N can be seen as balancing between optimizing for all orders vs the exact denoising order and which one is more beneficial for performance.

**Questions:**

1. Could you elaborate more on the differences between concurrent work. TraceRL and your work? As traceRL also merges timesteps for trajectory likelihood evaluation.
2. Regarding Algo1's implementation: would it be more computationally efficient to compute and collect likelihood estimates for all N segments first, then perform a single backward pass over the accumulated gradients? Or is computing one segment, immediately backpropagating, then moving to the next—more efficient? The could have memory and optimization implications.
3. When you compute and collect trajectory likelihoods during training after generation, do you store and reuse the exact denoising order from the generation phase, or do you randomly sample a new denoising order for each gradient update?

---

### Official Review · Reviewer_bkFN · 2025-11-08

**Soundness:** 2
**Presentation:** 2
**Contribution:** 2
**Rating:** 2
**Confidence:** 5

**Summary:**

The paper focuses on post-training/RLHF on diffusion language models (dllm), and more specifically aimed at improving reasoning performance. The main challenge is the lack of efficient likelihood computation for dllms. To address this the authors propose two likelihood estimators: d2-StepMerge, which trades computational cost for bias by evaluating on a subset of time-steps, and d2-AnyOrder, which allows estimation given an “any-order causal” architecture. The paper compares performance on logical and mathematical reasoning benchmarks for DLMs, using 4 different benchmarks; and show improved results than previous models under equal FLOP budgets (for d2-StepMerge). They also analyze performance for d2-AnyOrder on text-modeling task.

**Strengths:**

The paper directly tackles a challenging problem: applying PGRL methods to DLMs. The characterization of the "any-order causality" property is a good conceptual framing for architecture design. The paper is generally well-structured and generally readable.
The d2-stepMerge is a generalization of the d1-method.

**Weaknesses:**

The improved results largely stem from the performance on sudoku. On other tasks the improvement is small. Is there any specific reason for this. Additionally while the accuracy-flop comparison is on sudoku ( primarily because the difference is bigger there), but what about other tasks.

The paper mentions diffucoder's attempt to reduce vairance in likelihood estimation. They similarly also consider wd1 which also is focused on a similar problem. But they do not present results on these tasks from diffucoder's coupled loss. Since that seems computationally less expensive, that should be considered as a baseline. There are other works which also address the same problem [1,2] which should be discussed (and compared against if feasible)

Why was d2-anyorder not tried on the reasoning tasks. Additionally, the paper suggests that MDLM losses should be sufficient for any-order but then says current models are not. I do not understand why? Additionally presuming it is true, there needs to be greater and seperate analysis of the 'anyorder' estimate. Finally, a reasonable ablation would be to do fine-tuning with d2-anyorder on llada/d1 to compare performance. Given the title being "d2" but using two different estimators on two completely different models/tasks makes it seem that the second estimator was shoe-horned in.

d2-stepmerge is a pretty natural and obvious extension of d1/diffu-GRPO. The theoretical analysis is also pretty straighforward and not novel. The proofs around d2-anyorder are also not precise enough (though i do believe the results are correct).

1 Test-Time Alignment of Discrete Diffusion Models with Sequential Monte Carlo
2 PADRE: Pseudo-Likelihood based Alignment of Diffusion Language Models

**Questions:**

I have raised some questions in weaknesses earlier. Some additional questions/suggestions:

Can the authors provide the empirical relation between the number of StepMerge partitions (N) and the bias in likelihood estimation.
Additionally more detailed quantitative comparison on non-sudoku tasks would be good

What specific changes are required to instill anyorder property in a model like LLaDA? Is it feasible to retrofit this property into a pre-trained model, or does it require training a new model from scratch?

Shouldn't the comparison with DDPO (for d2-anyorder) be one which involved the KL penalty. It seems odd to remove KL penalty from your post-training algorithm for comparison than the other way around?

---

### Meta-Review · Area_Chair_UuG1 · 2026-01-06

**Summary:**

This paper proposes a novel reinforcement learning algorithm for improving the performance of diffusion language models at reasoning tasks.

**Reviewer Concerns:**

The reviewers had a wide variety of concerns; the most notable concerns shared across reviewers were: (1) the confusing scope of contributions, with two algorithms d2-stepmerge and d2-anyorder that appear unrelated, and (2) the scope of the experimental comparisons. For (1), the main concerns were that d2-stepmerge is not that novel, but d2-anyorder is not comprehensively evaluated due to its limited applicability. For (2), the reviewers pointed out that most of the improvements came from Sudoku, a limited task, and that there are missing baselines for comparison.

**Reviewer Scores:**

The authors did not provide a response, so the scores would not change.

---

### Decision · Program_Chairs · 2026-01-26

Reject